# Unsupervised Trajectory Optimization for 3D Registration in Serial Section Electron Microscopy using Neural ODEs

**Zhenbang Zhang**[*]
MBZUAI; Shandong University
zhangzhenbang2021@gmail.com

**Jingtong Feng**[*]
Shandong University
jingtongf404@gmail.com

**Hongjia Li**
Beijing Institute of Technology
lihongjia96@gmail.com

**Haythem El-Messiry**
Canadian University Dubai
haythem.elmessiry@cud.ac.ae

**Zhiqiang Xu**[†]
MBZUAI
zhiqiang.xu@mbzuai.ac.ae

**Renmin Han**[†]
Shandong University
hanrenmin@sdu.edu.cn

## Abstract

Series Section Electron Microscopy (ssEM) has emerged as a pivotal technology for deciphering nanoscale biological architectures. Three-dimensional (3D) registration is a critical step in ssEM, tasked with rectifying axial misalignments and nonlinear distortions introduced during serial sectioning. The core scientific challenge lies in achieving distortion mitigation without erasing the natural morphological deformations of biological tissues, thereby enabling faithful reconstruction of 3D ultrastructural organization. In this study, we present a paradigm-shifting optimization framework that rethinks 3D registration through the lens of manifold trajectory optimization. We propose the first continuous trajectory dynamics formulation for 3D registration and introduce a novel optimization strategy. Specifically, we introduce a dual optimization objective that inherently balances global trajectory smoothness with local structural preservation, while developing a solver that combines Gauss-Seidel iteration with Neural ODEs to systematically integrate biophysical priors with data-driven deformation compensation. Extensive experiments on multiple datasets spanning diverse tissue types demonstrate our method's superior performance in structural restoration accuracy and cross-tissue robustness.

## 1 Introduction

Series Section Electron Microscopy (ssEM) has emerged as a powerful technology for nanoscale Three-dimensional (3D) visualization of biological systems. Its impact spans diverse domains including neuroscience connectomics [84, 79], developmental biology [19, 43], and clinical diagnostics [29]. Recently, MICrONS project [1, 13, 15, 4] utilizes ssEM as the fundamental method for neural circuit reconstruction. The ssEM workflow typically involves a series of computational steps, including two-dimensional (2D) stitching [21], 3D registration [53, 82], and 3D segmentation [40, 47].

---

[*]Co-First Authors.
[†]Corresponding Authors.

39th Conference on Neural Information Processing Systems (NeurIPS 2025).

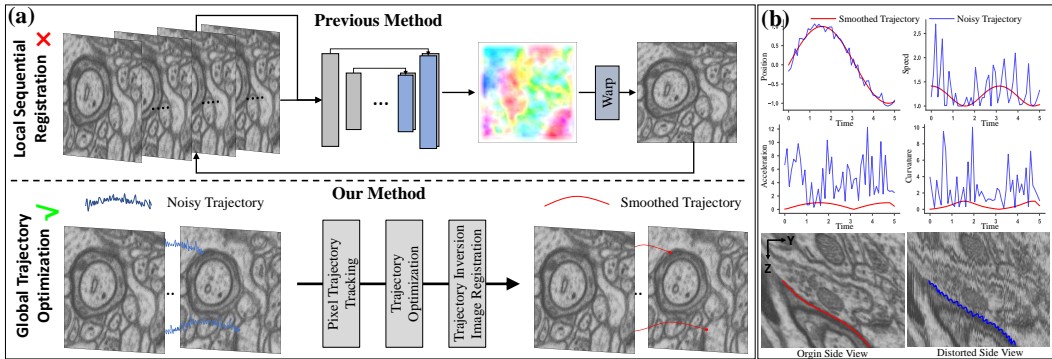

Figure 1: **Trajectory analysis and inspiration:** Left: Previous methods is limited to adjacent registration, while our method achieves a paradigm shift by global trajectory optimization. Right: The upper-right demonstrates **position/velocity/acceleration/curvature** comparisons between smoothed and noisy 1D trajectories. The lower-right displays the side view of the original and distorted data. This inspires us to explore the inner link between trajectory curvature mutation and nonlinear distortion from the perspective of biophysical motion.

Among these, 3D registration is particularly critical, as it corrects axial misalignments and nonlinear distortions, ultimately determining the accuracy and reliability of the final 3D structure.

The fundamental challenge stems from the topological entanglement of biological signal and technical noise – specifically, the indistinguishability between natural cellular morphological deformation and nonlinear distortion induced by sample preparation [76, 51]. Furthermore, state-of-the-art equipment now generates TB-scale image data daily [77]. The massive data volume and error accumulation in long sequences [36] greatly complicate fast and accurate 3D registration.

Several effective methods have been developed [56, 80, 86, 76, 42, 82]. However, these methods often fail to account for natural deformations between slices, and prioritizing pixel-level similarity may erase biologically meaningful changes, compromising the reconstructed structure. Recently, Zhang et al. [82] decouple natural deformation from nonlinear distortion by modeling them as low and high-frequency components. This insight motivated us to investigate the intrinsic characteristics of nonlinear distortion through the lens of biological motion coherence. We discovered that nonlinear distortion is essentially a sudden change in local curvature. As shown in Figure 1(b), anatomical structures inherently exhibit smooth manifold properties, maintaining evolutionary consistency in morphological changes. Mechanical sectioning, however, disrupts this continuity, introducing abrupt curvature mutations that contradict tissue biomechanics.

Building upon this, we propose a noval optimization framework, NeuroTrajReg, which redefines 3D registration as a motion trajectory optimization problem. We establish a continuous trajectory dynamics formulation for ssEM, integrating spatiotemporal constraints to effectively eliminate nonlinear distortions while preserving the natural evolution of tissue morphology. We introduce a dual optimization objective that inherently balances global trajectory smoothness and local structural preservation, while developing a solver that combines Gauss-Seidel iteration with Neural ODEs to systematically integrate biophysical priors with data-driven deformation compensation. Extensive experiments spanning diverse tissue types demonstrate NeuroTrajReg's superior performance in structural restoration accuracy and robustness across different tissue types. The main contributions of this paper are summarized as follows:

- We analyze the nonlinear distortions from the perspective of biological motion coherence. Based on this analysis, we redefine 3D registration as a continuous manifold trajectory optimization problem and propose the first continuous trajectory dynamics formulation.

- We propose a novel framework that integrates biophysical priors with data-driven deformation compensation, achieving a theory-guided yet data-corrected approach.

- Extensive experiments demonstrate that NeuroTrajReg achieves superior performance in accurately and faithfully reconstructing biological 3D structures while maintaining cross-tissue robustness.

## 2 Related works

**3D registration for series section electron microscopy.** Several software packages have been developed for 3D registration [39, 38, 66]. Among them, TrakEM2 [56] is one of the most widely used tools, performing 3D registration by iteratively optimizing a spring-connected particle system. Recently, deep learning-based methods have emerged as promising alternatives for 3D registration [80, 86, 42]. SEAMLeSS [53] improves robustness through vector voting and achieves global registration via combined attenuation transformations. Recently, Zhang et al. [82] proposed a Gaussian filter-based 3D registration method that approaches the problem from a frequency-domain perspective.

**Medical image registration.** Medical image registration [85, 48, 24, 16], closely related to 3D registration, has achieved notable progress [85, 52, 5, 83, 20, 22]. Existing methods are typically categorized by their transformation models. Dense models [2, 9, 37, 8] estimate voxel-wise mappings to form dense deformation fields, whereas interpolation-based models [30, 70, 58, 61] approximate deformations using basis functions (e.g., B-splines) over spatial grids. Despite their success, these approaches mainly address image pairs. In contrast, 3D registration demands handling nonlinear distortions while maintaining axial continuity.

**Neural ordinary differential equations.** Neural Ordinary Differential Equations (Neural ODEs) [6] integrate differential equation solvers into neural networks, enabling continuous representations with adaptive computation and enhanced parameter efficiency. Formally, given hidden state $\mathbf{h}(t)$, a Neural ODEs defines its dynamics through a parameterized function $f_\theta$:

$$\frac{d\mathbf{h}(t)}{dt} = f_\theta(\mathbf{h}(t), t), \tag{1}$$

and the final state is obtained by solving $\mathbf{h}(t_1) = \mathbf{h}(t_0) + \int_{t_0}^{t_1} f_\theta(\mathbf{h}(t), t)dt$. Numerous variants of Neural ODEs have emerged to capture more complex transformations [57, 75, 49, 44, 35, 27]. Neural ODEs have been widely applied across various fields [50, 31, 55, 68]: Vid-ODE [50] enables continuous-time video synthesis, NODEO [74] adapts it for deformable image registration, and Latent ODE [55] models continuous-time sequences.

**Reference line smoothing algorithms.** In autonomous driving [81, 65], the reference line smoothing algorithm [12, 60, 17] geometrically optimizes global paths to generate smooth trajectories with curvature continuity, kinematic feasibility, and minimal deviation from the original reference. Its primary role involves eliminating curvature discontinuities between discrete waypoints, constraining maximum curvature within vehicle kinematic limits. The loss function is formulated as:

$$J = \underbrace{\sum_{i=2}^{n-1} \|\mathbf{p}_{i-1} - 2\mathbf{p}_i + \mathbf{p}_{i+1}\|^2}_{\text{Smoothness}} + \lambda \underbrace{\sum_{i=1}^{n} \|\mathbf{p}_i - \mathbf{q}_i\|^2}_{\text{Fidelity}} + \underbrace{\sum_{i=1}^{n} \delta(|\kappa_i| \leq \kappa_{\max})}_{\text{Constraints}}, \tag{2}$$

where the first term penalizes curvature variation via second-order differencing, the second enforces proximity to original waypoints $\mathbf{q}_i$ with adaptive weight $\lambda$, and the third applies indicator function $\delta(\cdot)$ to enforce maximum curvature $\kappa_{\max}$. Optimization strategies include convex quadratic programming (QP) for real-time computation, sequential QP (SQP) for nonlinear constraint handling [12, 17], spline curves [72, 14], and mass-spring physical simulation [34, 23].

## 3 Proposed method: NeuroTrajReg

For the problem of 3D registration, we propose a dynamics-based nonlinear distortion correction framework for ssEM image sequences. Given a stack of microscopic images $\{I_z\}_{z=0}^{N-1}$ affected by complex nonlinear distortions, our objective is to reconstruct a 3D structure that faithfully preserves the biological morphology by establishing a registration model with biophysical priors. The core challenge lies in dynamically balancing two conflicting demands: on the one hand, eliminating nonlinear spatial distortions induced by external factors, and on the other hand, preserving the intrinsic morphological characteristics of the biological specimen, such as tissue deformation and developmental topological changes.

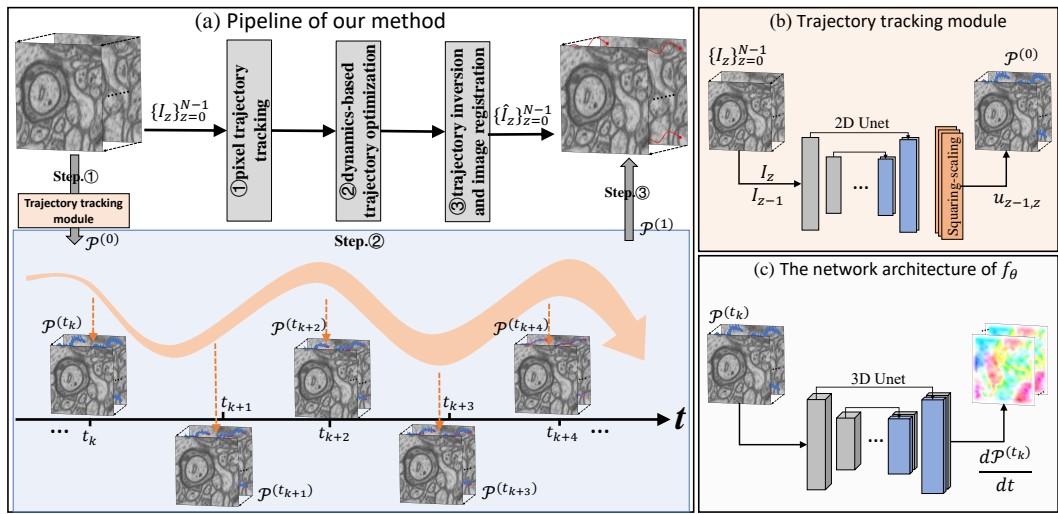

Figure 2: (a) The overall pipeline of NeuroTrajReg. Our approach can be divided into three main components: 1) Pixel trajectory tracking, 2) Dynamics-based trajectory optimization, and 3) Trajectory inversion and image registration. (b) Illustration of the trajectory tracking module. (c) The network architecture of $f_\theta$ (Neural ODEs) in dynamics-based trajectory optimization module.

## 3.1 Problem formulation

In this section, we introduce the concept of pixel-wise motion trajectory tracking and smoothing. Drawing inspiration from the classical Laplace equation [7] and the reference line smoothing algorithms [12], we formulate the 3D registration as a continuous manifold trajectory optimization problem. Specifically, each pixel $p_i \in \mathbb{R}^2$ is treated as a particle within a continuum medium, whose displacement along the slice axis $z \in [z_0, z_{N-1}]$ generates a parameterized trajectory $\mathcal{P}_i$. While this trajectory ideally follows natural morphological variations, nonlinear distortions induce local curvature abruptions. To address this, we establish spatiotemporal smoothness constraints on pixel trajectories, transforming traditional 3D registration into a physically interpretable continuous trajectory optimization problem. NeuroTrajReg is grounded in three key principles: 1) physical motion coherence in biological tissues; 2) Laplace-based trajectory smoothing; 3) spatiotemporal continuity in microscopy imaging. We model pixel-wise motion from individual dynamics to global trajectory constraints with strict mathematical consistency.

**Single-particle dynamics.** Let $\mathcal{P}_i$ denote the parameterized trajectory. We formulate the single-particle optimization problem as follows:

$$\min_{\mathcal{P}_i} \left\{ \underbrace{\int_{z_0}^{z_{N-1}} \|\mathcal{P}_i(z) - \mathcal{P}_i^{(0)}(z)\|^2 dz}_{\text{Data fidelity}} + \lambda \underbrace{\int_{z_0}^{z_{N-1}} \|\nabla^2 \mathcal{P}_i(z)\|^2 dz}_{\text{Laplacian smoothness}} \right\}, \tag{3}$$

where $\mathcal{P}_i^{(0)}$ denotes the observed origin trajectory, $\nabla^2$ is the axial Laplacian operator, and $\lambda > 0$ controls the regularization strength. The Laplacian term enforces second-order temporal smoothness by penalizing acceleration discontinuities, effectively filtering biologically implausible instantaneous acceleration changes.

**Holistic trajectory constraint.** For image stack $\{I_z\}_{z=0}^{N-1}$, we formulate the holistic optimization:

$$\min_{\{\mathcal{P}_i\}} \sum_{i=1}^{M} \left[ \int_{z_0}^{z_{N-1}} \|\mathcal{P}_i(z) - \mathcal{P}_i^{(0)}(z)\|^2 dz + \lambda \int_{z_0}^{z_{N-1}} \|\nabla^2 \mathcal{P}_i(z)\|^2 dz + \mu S_i \right], \tag{4}$$

where $M$ is the total number of trajectories, corresponding to the number of pixels in image $I_z$, and the spatial coherence term $S_i$ is defined as:

$$S_i := \sum_{j \in \mathcal{N}(i)} \int_{z_0}^{z_{N-1}} \|\mathcal{P}_i(z) - \mathcal{P}_j(z)\|^2 dz, \tag{5}$$

and weighted by $\mu > 0$ for preserving local topologies within each slice, where $\mathcal{N}(i)$ denotes the set of spatial neighbors of point $i$ within the same slice. This dual regularization strategy embeds two fundamental biological principles:

- **Temporal smoothness**. It preserves smooth and stable tissue morphology evolution by constraining axial acceleration discontinuities through Laplacian regularization, preventing unreliable instantaneous trajectory changes.
- **Spatial consistency**. It maintains the local topology between slice pixels by ensuring the consistency of trajectory smoothing for neighboring pixels.

## 3.2 Trajectory optimization for 3D registration

Our approach, as shown in Figure 2(a), mainly consists of three components: 1) pixel trajectory tracking, 2) dynamics-based trajectory optimization, and 3) trajectory inversion and image registration. Specifically, we perform trajectory tracking for each pixel in the image stack. Then, we apply our proposed trajectory optimization algorithm to obtain smooth pixel trajectories. Afterward, we compute the trajectory displacements and invert them back to the pixel grid to generate a deformation field, which is then used to register the image stack. We will now elaborate on the implementation and contributions of each component.

**Pixel trajectory tracking.** Given image stack $\{I_z\}_{z=0}^{N-1} \in \mathbb{R}^{N \times 1 \times H \times W}$, our goal is to track the trajectory of each pixel over time, accounting for subpixel displacements. We use the network shown in Figure 2(b) for estimating the displacement field $u_{z-1,z} \in \mathbb{R}^{H \times W \times 2}$ in $x$-axis and $y$-axis between adjacent slices. To model the temporal trajectory of every pixel, we introduce a trajectory volume $\mathcal{P} \in \mathbb{R}^{N \times H \times W \times 2}$, where $\mathcal{P}(z)$ stores the spatial coordinate in slice $z$. This trajectory is initialized as:

$$\mathcal{P}(0, y, x) = (y^0, x^0), \tag{6}$$

indicating that the trajectory of each pixel starts from its position in the first slice. Here, $\mathcal{P}(z, y, x)$ denotes the displacement vector at index $(y, x)$ in slice $z$.

We recursively propagate the position using the displacement field from the previous slice. But the pixel location $\mathcal{P}(z-1, y, x)$ may not be aligned with the discrete grid of $u_{z-1,z}$, as accumulated displacements often lead to non-integer coordinates. As a result, bilinear interpolation of $u_{z-1,z}$ is required to accurately estimate the flow at subpixel positions. Formally, the trajectory propagation is governed by the transport equation:

$$\mathcal{P}(z, y, x) = \mathcal{P}(z-1, y, x) + \widetilde{u}_{z-1,z}(\mathcal{P}(z-1, y, x)), \tag{7}$$

where $\widetilde{u}_{z-1,z}(\cdot)$ denotes the interpolated displacement field.

**Dynamics-based trajectory optimization.** According to Eq. (3), we model trajectory smoothing as the weighted minimization of discrete curvature energy and deviations from the original trajectory. To solve this optimization objective, we employ the Gauss-Seidel method [18] to iteratively smooth the trajectory. The Gauss-Seidel method is a classical algorithm for solving linear systems. Its key characteristic lies in accelerating convergence by immediately incorporating the most recently updated values. The corresponding Gauss-Seidel iteration is given by:

$$\mathcal{P}^{(k+1)}(z) = \frac{\mathcal{P}^{(k+1)}(z-1) + \mathcal{P}^{(k)}(z+1) + \lambda \mathcal{P}^{(0)}(z)}{2 + \lambda}, \quad z = 1, \ldots, N-1. \tag{8}$$

with boundary values fixed as $\mathcal{P}^{(k)}(0) \equiv \mathcal{P}^{(0)}(0)$ and $\mathcal{P}^{(k)}(N-1) \equiv \mathcal{P}^{(0)}(N-1)$. In each iteration $k$, the internal positions $z$ are updated from the 1 to the $N-1$, where each $\mathcal{P}^{(k+1)}(z)$ is computed using the most recent value of its predecessor $\mathcal{P}^{(k+1)}(z-1)$ and the previous value of its successor $\mathcal{P}^{(k)}(z+1)$. A detailed derivation is provided in the supplementary material.

While the Gauss-Seidel method provides a theoretically grounded approach for solving linear systems through explicit point-wise updates with guaranteed convergence, its practical effectiveness is inherently limited by discrepancies between idealized theoretical models and real-world scenarios. Contrastingly, Neural ODEs [6] excel in learning continuous dynamical systems through implicit dynamic modeling, effectively capturing complex nonlinear deviations.

Given the exceptional capability of Neural ODEs in modeling continuous dynamic systems [69, 31, 46], we use Neural ODEs to reformulate trajectory smoothing as the dynamic evolution of a continuous trajectory. The network architecture of $f_\theta$ in Neural ODEs is shown in Figure 2(c). The trajectory constraints are enforced through integration of velocity field $\mathcal{V} = \{v_z\}_{z=1}^{N-1}$ satisfying:

$$\mathcal{V}^{(t)} = \frac{d\mathcal{P}^{(t)}}{dt} = f_\theta\left(\mathcal{P}^{(t)}, t\right), \quad \mathcal{P}^{(t_{k+1})} = \mathcal{P}^{(t_k)} + \int_{t_k}^{t_{k+1}} \mathcal{V}^{(t)} dt, \tag{9}$$

where $f_\theta$ is a neural network with trainable parameters $\theta$.

Inspired by Universal Differential Equations (UDEs) [54], we propose a dynamically adaptive optimization system that synergistically combines the explicit iterative convergence of Gauss-Seidel with the implicit dynamic modeling capabilities of Neural ODEs:

$$\mathcal{P}^{(t_N)} = (1-\alpha) \underbrace{\mathcal{G}(\mathcal{P}^{(t_0)})}_{\text{Gauss-Seidel core}} + \alpha \underbrace{\text{ODESolver}\big[f_\theta, \mathcal{P}^{(t_0)}, [t_0, t_N]\big]}_{\text{Neural compensator}}, \tag{10}$$

where $\mathcal{G}(\cdot)$ denotes the Gauss-Seidel operator, which takes the initial trajectory at time $t_0$ as input and outputs a smoothed trajectory at time $t_N$. ODESolver[·] represents the neural dynamic correction over time interval $[t_0, t_N]$. The adaptive weight $\alpha \in [0, 1]$ evolves through:

$$\alpha = \sigma\left(\beta \cdot \left(1 - \frac{\Delta^{(t_N)}}{\Delta^{(t_0)}}\right)\right), \tag{11}$$

where $\sigma(\cdot)$ denotes the sigmoid activation ensuring smooth transitions, $\beta$ controls the adaptation rate sensitivity, and $\Delta^{(t_k)} = \|\mathcal{G}(\mathcal{P}^{(t_k)}) - \mathcal{P}^{(t_k)}\|$ quantifies theoretical operator progress.

Therefore, our architecture follows a *theory-guided, data-corrected approach*, incorporating an adaptive blending module that automatically balances theoretical components and corrective terms. It takes advantage of Gauss-Seidel for rapid initial convergence, ensuring numerical stability and accelerating optimization. Neural ODEs are then employed to capture higher-order nonlinear effects and compensate for the gaps between the model and reality.

**Trajectory inversion and image registration.** After trajectory optimization, we obtain the resulting trajectories $\mathcal{P}^{(t_N)} \in \mathbb{R}^{N \times H \times W \times 2}$, where each element $\mathcal{P}_i^{(t_N)}(z) \in \mathbb{R}^2$ represents the evolved coordinate of the $i$-th trajectory $\mathcal{P}_i^{(t_N)}$ at slice $z$. Based on this, the displacement $\phi \in \mathbb{R}^{N \times H \times W \times 2}$ can be computed by comparing it with the observed original trajectory $\mathcal{P}^{(0)}$:

$$\phi = \mathcal{P}^{(t_N)} - \mathcal{P}^{(0)}. \tag{12}$$

The displacement $\phi$ encodes the offset information of $\mathcal{P}^{(t_N)}$. However, it is important to note that this does not exactly correspond to the displacement of pixel grid points, as the coordinates in $\mathcal{P}^{(t_N)}$ are not necessarily integers. Therefore, inspired by the surface splatting [87, 33] in computer graphics, we propose an efficient bilinear splatting to invert the trajectories and obtain accurate displacements at pixel grid locations (detailed in the supplementary material):

$$\{\Phi_z\}_{z=0}^{N-1} = \mathcal{I}_{\text{bs}}(\mathcal{P}^{(t_N)}, \phi), \tag{13}$$

where $\mathcal{I}_{\text{bs}}$ means bilinear splatting and $\{\Phi_z\}_{z=0}^{N-1} \in \mathbb{R}^{N \times H \times W \times 2}$ represents the deformation fields corresponding to the image stack $\{I_z\}_{z=0}^{N-1}$. These deformation fields are subsequently used to register the image stack:

$$I_z = \Phi_z \circ I_z, \quad z = 0, \ldots, N-1, \tag{14}$$

where $\circ$ denotes the warping operation.

## 3.3 Loss functions

The trajectory tracking network is optimized using a composite unsupervised loss function comprising a normalized cross-correlation (NCC) data fidelity term and a diffusion regularizer applied to the displacement fields, with balancing coefficient $\lambda$. The loss function is formally expressed as:

$$\mathcal{L}_{\text{traj}} = \text{NCC}\left(I_z, I_{z-1} \circ u_{z-1,z}\right) + \lambda \|\nabla u_{z-1,z}\|_2^2, \tag{15}$$

where ∘ indicates the spatial warping operator, and $\nabla$ computes first-order spatial gradients via finite difference approximation.

According to Eq. (4), the Neural ODEs training objective is formulated as a composite unsupervised loss function comprising three components:

$$\mathcal{L} = \sum_{i=1}^{M} \Big( \underbrace{\sum_{z=z_0}^{z_{N-1}} \|\mathcal{P}_i(z) - \mathcal{P}_i^{(0)}(z)\|^2}_{\text{Data fidelity term}} + \lambda \underbrace{\sum_{z=z_0}^{z_{N-1}} \|\nabla^2 \mathcal{P}_i(z)\|^2}_{\text{Trajectory smoothness term}} + \mu \underbrace{\sum_{z=z_0}^{z_{N-1}} \sum_{j \in \mathcal{N}(i)} \|\mathcal{P}_i(z) - \mathcal{P}_j(z))\|^2}_{\text{Spatial consistency term}} \Big),$$

(16)

where $\mathcal{P}_i^{(0)}$ denotes the observed origin trajectory. The trajectory smoothness term, weighted by $\lambda > 0$, penalizes acceleration discontinuities through the temporal Laplacian operator $\nabla^2$, discretized with second-order differences. The spatial consistency term, weighted by $\mu > 0$, preserves local topological relationships within each slice.

# 4 Experiments

## 4.1 Datasets

We evaluate our algorithm on six publicly available datasets from the OpenOrganelle platform [77], covering a variety of mouse tissues such as heart, kidney, liver, skin, and pancreas. These allow for a comprehensive evaluation of NeuroTrajReg's applicability and robustness across diverse tissue types. For real-world data, we utilized the female fruit fly brain neural dataset (FemFlyBrain) [63]. More details can be found in the supplementary material.

## 4.2 Implementation details

We implement our method using PyTorch, and all experiments are conducted on an NVIDIA A800 GPU with 80GB of memory. For the pixel trajectory tracking, we employ a 2D U-Net architecture to capture the displacements between adjacent slices. Neural ODEs employ a 3D U-Net architecture to parameterize $f_\theta$. The network takes pixel trajectories $\mathcal{P}$ as input and generates a time-dependent velocity field $\mathcal{V}$. More details can be found in the supplementary material.

## 4.3 Results

In Table 1, we compare NeuroTrajReg with advanced 3D registration techniques, including EMReg [42], EFSR [76], TrakEM2 [56], SEMLeSS [53], and GaussReg [82], using synthetic datasets spanning six diverse tissue types. Our approach, along with the supervised GaussReg method, demonstrates superior performance due to consideration of natural deformations, which other methods typically overlook. Methods overly focused on pixel-level similarity can neglect biologically relevant movements such as cellular dynamics, as shown in Figure 7. Moreover, compared to the supervised-manner GaussReg, our method uniquely benefits from global trajectory optimization, enabling the computation of a single interpolation deformation across slices. Conversely, GaussReg is constrained by a local receptive field, necessitating multiple interpolation steps. For the performance on real-world datasets, we selected several volumetric samples from the FemFlyBrain dataset [63]. The registration results are visualized in Figure 5, where we employed the interpolation and visualization tools provided by FIJI [59] to display the side views of the image stacks. Although GaussReg reduces nonlinear distortions, it also erroneously removes some structural textures, leading to a noticeable deviation from the original data. In contrast, our method not only effectively corrects the nonlinear distortions in the image stacks but also better preserves local structural details.

To further evaluate our approach, Figure 3 compares the error accumulation across approximately 1000 slices for SEMLeSS [53], GaussReg [82], and our method. SEMLeSS suffers significantly from error accumulation over long sequences. Although GaussReg performs reasonably well, it exhibits lower precision and higher variance compared to our method, indicating less stable performance. Our method achieves superior average SSIM accuracy and demonstrates substantially greater stability , maintaining consistent performance without noticeable degradation even after 1000 slices. Additional results are provided in the supplementary material.

Table 1: The performance of different 3D registration methods in six synthetic datasets [77].

| | | Mus Heart | Mus Kidney | Mus Liver-3 | Mus Liver | Mus Pancreas | Mus Skin |
|---|---|---|---|---|---|---|---|
| EMReg [47] | MI | $0.73_{\pm 0.14}$ | $0.65_{\pm 0.16}$ | $0.89_{\pm 0.17}$ | $0.79_{\pm 0.18}$ | $0.76_{\pm 0.14}$ | $0.66_{\pm 0.18}$ |
| | SSIM | $0.55_{\pm 0.03}$ | $0.44_{\pm 0.04}$ | $0.52_{\pm 0.04}$ | $0.43_{\pm 0.03}$ | $0.43_{\pm 0.02}$ | $0.50_{\pm 0.07}$ |
| | NCC | $0.90_{\pm 0.03}$ | $0.81_{\pm 0.05}$ | $0.86_{\pm 0.05}$ | $0.78_{\pm 0.05}$ | $0.81_{\pm 0.03}$ | $0.87_{\pm 0.04}$ |
| EFSR [76] | MI | $1.15_{\pm 0.10}$ | $1.26_{\pm 0.15}$ | $1.45_{\pm 0.14}$ | $1.36_{\pm 0.16}$ | $1.33_{\pm 0.13}$ | $1.03_{\pm 0.20}$ |
| | SSIM | $0.67_{\pm 0.04}$ | $0.63_{\pm 0.05}$ | $0.65_{\pm 0.04}$ | $0.56_{\pm 0.04}$ | $0.57_{\pm 0.04}$ | $0.60_{\pm 0.07}$ |
| | NCC | $0.98_{\pm 0.00}$ | $0.97_{\pm 0.01}$ | $0.97_{\pm 0.00}$ | $0.95_{\pm 0.01}$ | $0.95_{\pm 0.01}$ | $0.97_{\pm 0.01}$ |
| TrakEM2 [56] | MI | $1.01_{\pm 0.03}$ | $1.24_{\pm 0.10}$ | $1.48_{\pm 0.07}$ | $1.47_{\pm 0.08}$ | $1.48_{\pm 0.06}$ | $1.08_{\pm 0.15}$ |
| | SSIM | $0.57_{\pm 0.02}$ | $0.61_{\pm 0.03}$ | $0.66_{\pm 0.03}$ | $0.61_{\pm 0.03}$ | $0.65_{\pm 0.03}$ | $0.64_{\pm 0.06}$ |
| | NCC | $0.98_{\pm 0.00}$ | $0.97_{\pm 0.00}$ | $0.98_{\pm 0.00}$ | $0.97_{\pm 0.00}$ | $0.97_{\pm 0.00}$ | $0.98_{\pm 0.01}$ |
| SEMLeSS [53] | MI | $1.00_{\pm 0.12}$ | $1.07_{\pm 0.14}$ | $1.35_{\pm 0.17}$ | $1.29_{\pm 0.19}$ | $1.18_{\pm 0.15}$ | $0.89_{\pm 0.18}$ |
| | SSIM | $0.58_{\pm 0.04}$ | $0.52_{\pm 0.04}$ | $0.61_{\pm 0.05}$ | $0.52_{\pm 0.06}$ | $0.50_{\pm 0.05}$ | $0.51_{\pm 0.07}$ |
| | NCC | $0.97_{\pm 0.01}$ | $0.96_{\pm 0.01}$ | $0.97_{\pm 0.01}$ | $0.95_{\pm 0.01}$ | $0.95_{\pm 0.01}$ | $0.96_{\pm 0.01}$ |
| GaussReg [82] | MI | $\underline{1.43}_{\pm 0.05}$ | $\underline{1.54}_{\pm 0.11}$ | $\underline{1.83}_{\pm 0.07}$ | $\underline{1.83}_{\pm 0.08}$ | $\underline{1.76}_{\pm 0.06}$ | $\underline{1.40}_{\pm 0.16}$ |
| | SSIM | $\underline{0.83}_{\pm 0.02}$ | $\underline{0.78}_{\pm 0.03}$ | $\underline{0.82}_{\pm 0.02}$ | $\underline{0.80}_{\pm 0.02}$ | $\underline{0.80}_{\pm 0.02}$ | $\underline{0.82}_{\pm 0.03}$ |
| | NCC | $\mathbf{0.99}_{\pm 0.00}$ | $\mathbf{0.99}_{\pm 0.00}$ | $\mathbf{0.99}_{\pm 0.00}$ | $\mathbf{0.99}_{\pm 0.00}$ | $\mathbf{0.99}_{\pm 0.00}$ | $\mathbf{0.99}_{\pm 0.00}$ |
| Ours | MI | $\mathbf{1.47}_{\pm 0.08}$ | $\mathbf{1.67}_{\pm 0.14}$ | $\mathbf{1.88}_{\pm 0.11}$ | $\mathbf{1.85}_{\pm 0.21}$ | $\mathbf{1.81}_{\pm 0.12}$ | $\mathbf{1.52}_{\pm 0.17}$ |
| | SSIM | $\mathbf{0.87}_{\pm 0.02}$ | $\mathbf{0.85}_{\pm 0.02}$ | $\mathbf{0.87}_{\pm 0.01}$ | $\mathbf{0.84}_{\pm 0.15}$ | $\mathbf{0.86}_{\pm 0.18}$ | $\mathbf{0.86}_{\pm 0.03}$ |
| | NCC | $\mathbf{0.99}_{\pm 0.00}$ | $\mathbf{0.99}_{\pm 0.00}$ | $\mathbf{0.99}_{\pm 0.00}$ | $\mathbf{0.99}_{\pm 0.00}$ | $\mathbf{0.99}_{\pm 0.00}$ | $\mathbf{0.99}_{\pm 0.00}$ |

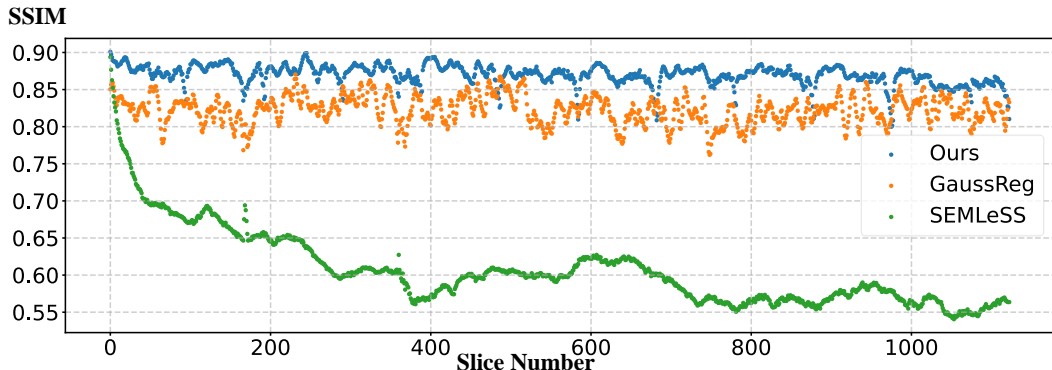

Figure 3: Error accumulation comparison on Mus Liver-3 dataset.

## 4.4 Ablation study

**Loss Terms.** We tested the impact of different components of the loss function (Eq. 16) on the results. The results are shown in Table 2, where TS, DF, and SC correspond to the Trajectory Smoothness term, Data Fidelity term, and Spatial Consistency term, respectively. Removing the Data Fidelity term decreases SSIM, as this leads to the trajectory deviating excessively from the original, with a risk of over-smoothing. Removing the Spatial Consistency term results in a decrease in SSIM, as failing to preserve local topology can lead to local pixel distortions. As for the TS term, it is key to trajectory smoothing, as it imposes constraints on the trajectory's acceleration, preventing unreasonable instantaneous variations. In conclusion, all components are essential and work together to ensure optimal performance.

**Hyperparameters.** We tested the impact of different hyperparameters $\lambda$ and $\mu$ in Eq. (16) on registration accuracy. $\lambda$ controls the smoothness of the trajectory to filter unreasonable instantaneous velocity changes, while $\mu$ regulates the local consistency of the displacement field to ensure topological consistency before and after registration. $\lambda$ has an important impact on registration performance; if it is too small, it fails to eliminate noise completely, while if it is too large, it may cause excessive deviation of the trajectory. In contrast, $\mu$ is more robust and has a lesser effect on registration performance.

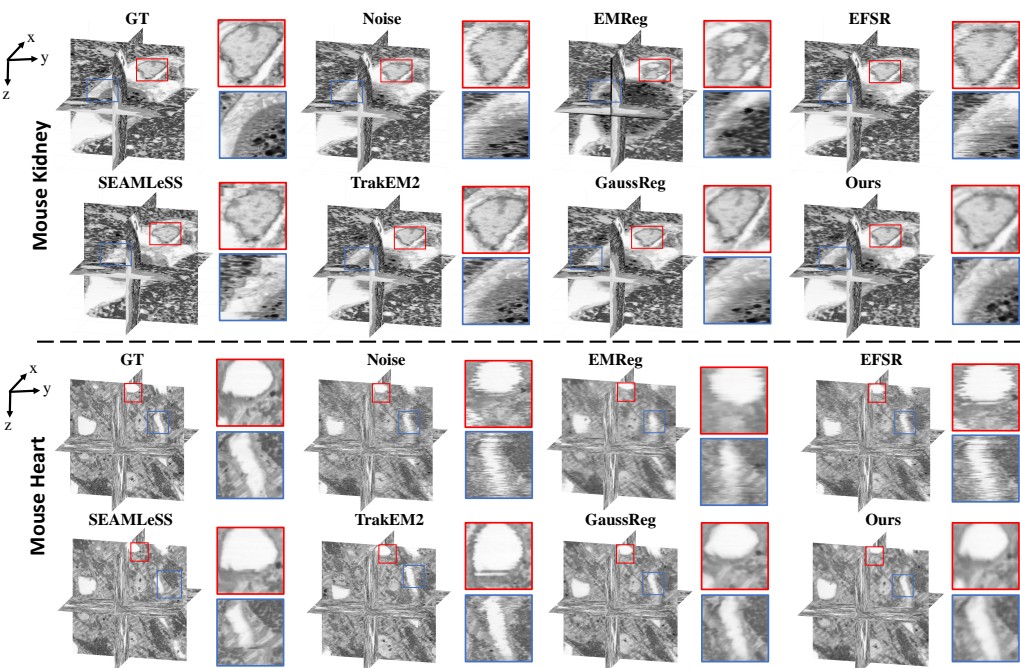

Figure 4: 3D visualization of registration results on the Mus Kidney and Mus Heart datasets.

Table 2: The ablation study of different loss terms.

| | TS Loss term | DF Loss term | SC Loss term | SSIM |
|---|---|---|---|---|
| | ✔ | ✘ | ✘ | 0.721 |
| Mus Heart | ✔ | ✔ | ✘ | 0.855 |
| | ✔ | ✘ | ✔ | 0.793 |
| | ✔ | ✔ | ✔ | 0.872 |
| | ✔ | ✘ | ✘ | 0.698 |
| Mus Kidney | ✔ | ✔ | ✘ | 0.822 |
| | ✔ | ✘ | ✔ | 0.774 |
| | ✔ | ✔ | ✔ | 0.851 |

Table 3: The ablation study of different hyper-parameter in the loss.

| SSIM | $\lambda\backslash\mu$ | 0.0001 | 0.001 | 0.01 | 0.1 |
|---|---|---|---|---|---|
| | 0.001 | 0.775 | 0.782 | 0.771 | 0.762 |
| Mus Heart | 0.01 | 0.795 | 0.806 | 0.783 | 0.779 |
| | 0.1 | 0.852 | 0.872 | 0.841 | 0.822 |
| | 1 | 0.811 | 0.819 | 0.805 | 0.787 |
| | 0.001 | 0.754 | 0.762 | 0.752 | 0.741 |
| Mus Kidney | 0.01 | 0.781 | 0.794 | 0.772 | 0.766 |
| | 0.1 | 0.833 | 0.851 | 0.821 | 0.815 |
| | 1 | 0.768 | 0.773 | 0.758 | 0.743 |

Table 4: Ablation study on the number of iterations in the Gauss-Seidel method.

| SSIM\Iterations | 50 | 100 | 150 | 200 | 250 | 300 |
|---|---|---|---|---|---|---|
| Mus Heart | 0.719 | 0.802 | 0.855 | 0.872 | 0.874 | 0.874 |
| Mus Kidney | 0.713 | 0.796 | 0.828 | 0.851 | 0.851 | 0.851 |
| Mus Liver | 0.695 | 0.774 | 0.811 | 0.842 | 0.844 | 0.844 |
| Mus Skin | 0.682 | 0.788 | 0.842 | 0.864 | 0.868 | 0.866 |

**Number of Iterations.** Table 4 investigates the effect of varying the number of iterations in the Gauss-Seidel method. From 0 to 200 iterations, the registration performance improves significantly and then gradually converges. Considering the trade-off between accuracy and efficiency, we select 200 iterations in practice to achieve a good balance.

**Module Analysis.** Our method can be decomposed into two main modules: the Gauss-Seidel method and Neural ODEs. The Gauss-Seidel method achieves fast convergence through iterative updates, while Neural ODEs effectively capture complex nonlinear distortions by modeling implicit dynamics. As shown in Table 5, we analyze the contribution of these key components in the trajectory smoothing scheme, and the results demonstrate the necessity and effectiveness of both modules.

**Different Solver.** Table 6 further studies the effect of different solvers for Neural ODEs. To avoid additional computational overhead, we adopt the Euler solver to balance performance and computational cost. More ablation experiments are provided in the supplemenatry material.

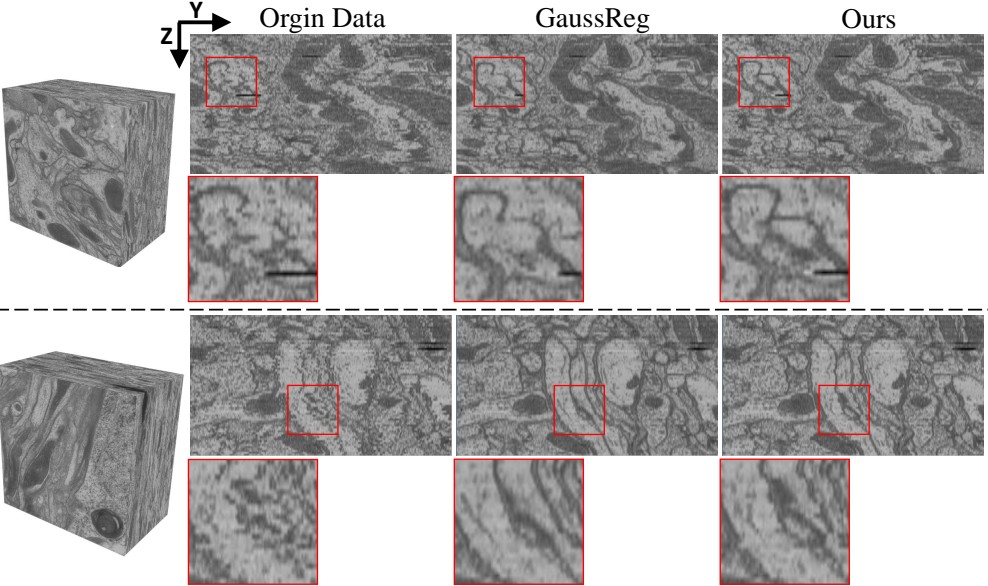

Figure 5: Side views of the original data, the registration results of GaussReg [82], and our method on two sampled volumes from the FemFlyBrain dataset [63].

Table 5: Ablation study of the components in our method.

|  | Gauss-Seidel | Neural ODEs | SSIM |
|---|---|---|---|
| | ✔ | ✘ | 0.847 |
| Mus Heart | ✘ | ✔ | 0.835 |
| | ✔ | ✔ | 0.872 |
| | ✔ | ✘ | 0.833 |
| Mus Kidney | ✘ | ✔ | 0.827 |
| | ✔ | ✔ | 0.851 |

Table 6: Ablation study of different solvers for Neural ODEs.

|  | Euler | | Dopri5 | | RK4 | |
|---|---|---|---|---|---|---|
| | Time | SSIM | Time | SSIM | Time | SSIM |
| Mus Heart | 2.45 | 0.872 | 8.33 | 0.877 | 7.68 | 0.876 |
| Mus Kidney | 2.41 | 0.851 | 8.52 | 0.855 | 7.66 | 0.854 |
| Mus Liver | 2.5 | 0.842 | 8.71 | 0.848 | 7.72 | 0.847 |
| Mus Skin | 2.46 | 0.864 | 8.39 | 0.869 | 7.25 | 0.868 |

## 5  Conclusion

In this paper, we present a noval 3D registration method for series section electron microscopy. We analyze the inherent characteristics of nonlinear distortion from the perspective of biological motion coherence, and reconsider 3D registration from the viewpoint of trajectory optimization. We introduce a dual optimization objective that balances global trajectory smoothness and local structural preservation, and we develop a solver that combines Gauss-Seidel iteration with Neural ODEs. Extensive experiments demonstrate that our method excels in structural restoration accuracy and cross-tissue robustness.

## 6  Acknowledgements

This research was supported by the National Key Research and Development Program of China [2021YFF0704300], Dubai Future Foundation (Award No. 2024CANAD-MES-061), and the Natural Science Foundation of Shandong Province [ZR2023YQ057].

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

# Appendix

## A1 Dataset Details and Metrics

### A1.1 Dataset Details

We selected six publicly available datasets from the OpenOrganelle platform [77] for simulation experiments. These datasets encompass high-resolution electron microscopy images of various mouse tissues, including the heart, kidney, liver, skin, and pancreas. The availability of ground-truth annotations in these datasets provides strong support for validating the applicability and robustness of our method across different types of biological tissues. For real-world data, we utilized the female fruit fly brain neural dataset (FemFlyBrain) [63]. These extensive datasets enabled us to validate the robustness of our method against real-world data.

**P7 Mouse Heart.** This dataset consists of heart tissue extracted from a wild-type C57BL/6J mouse at postnatal day 7. The experimental procedure followed IACUC guidelines for animal anesthesia. The tissue was fixed via perfusion with glutaraldehyde and sectioned using a vibratome. Subsequently, the sample underwent low-temperature reducing OTO staining, dehydration through a graded ethanol series, infiltration with Durcupan resin, and polymerization in an oven at 60°C.

**Mouse Kidney.** This dataset consists of kidney tissue extracted from an 8-week-old wild-type C57BL/6 mouse. The tissue was perfused with glutaraldehyde fixative and sectioned using a vibratome. After staining with reducing OTO at room temperature, the sample underwent dehydration, infiltration with a graded ethanol series, and Durcupan resin. Finally, the sample was polymerized in an oven at 60°C.

**P7 Mouse Liver.** This dataset comprises liver tissue collected from a wild-type, postnatal day 7 C57BL/6J mouse. Following anesthesia in accordance with IACUC guidelines, the sample was perfused with glutaraldehyde fixative and sectioned using a vibratome. After low-temperature reducing OTO staining, the tissue underwent dehydration, infiltration with graded ethanol and Durcupan resin, and was subsequently polymerized in a 60°C oven.

**Mouse Liver.** This dataset contains liver tissue from a wild-type, 8-week-old C57BL/6 mouse. The sample was fixed via perfusion with glutaraldehyde and sectioned using a vibratome. Subsequently, reducing OTO staining was performed at room temperature. The sample was then dehydrated, infiltrated with graded ethanol and Durcupan resin, and polymerized in a 60°C oven.

**P7 Mouse Skin.** This dataset includes skin tissue from a wild-type, postnatal day 7 C57BL/6J mouse. Mouse anesthesia was performed in accordance with IACUC guidelines, followed by perfusion fixation using glutaraldehyde and vibratome sectioning. The sample underwent low-temperature reducing OTO staining, dehydration, infiltration with graded ethanol and Durcupan resin, and was polymerized in a 60°C oven.

**P7 Mouse Pancreas.** This dataset consists of pancreas tissue from a wild-type, postnatal day 7 C57BL/6 mouse. The sample was fixed via perfusion with glutaraldehyde and sectioned using a vibratome. It underwent low-temperature reducing OTO staining, followed by dehydration, infiltration with graded ethanol and Durcupan resin, and polymerization in a 60°C oven.

**Female Fruit Fly Brain Neural Dataset.** FemFlyBrain [63] consists of the right hemisphere of a wild-type Oregon R female fruit fly brain. The brain was continuously sectioned at a thickness of 40 nanometers, covering regions including the medulla and downstream neuropils. The sections were imaged at a magnification of 35,000. The connectome within the medulla includes 379 neurons and 8,637 chemical synaptic contacts.

**Full Adult Fly Brain Dataset.** Full Adult Fly Brain Dataset (FAFB) [84] covers the entire depth of an adult fruit fly brain (approximately 250 $\mu$m). From the optimized sample, 7,062 consecutive sections of about 40 nm thickness were collected, optimized for high membrane contrast and fine ultrastructural preservation. A total of 7,050 sections (99.8%) were successfully imaged.

**Mouse Cortical Dataset.** The Mouse Cortical Dataset [32] provides a saturated reconstruction of a mouse neocortical sub-volume, in which all cellular elements (axons, dendrites, and glia) and various subcellular components (synapses, vesicles, spines, postsynaptic densities, and mitochondria) are fully annotated.

Table 7: Details of the synthetic datasets for testing phase.

| Datasets | Shape | Numbers | URL |
|---|---|---|---|
| P7 Mouse Heart | 1184×1184 | 1000 | https://openorganelle.janelia.org/datasets/jrc_mus-heart-1 |
| Mouse Kidney | 1184×1184 | 1000 | https://openorganelle.janelia.org/datasets/jrc_mus-kidney |
| P7 Mouse Liver | 1184×1184 | 1127 | https://openorganelle.janelia.org/datasets/jrc_mus-liver-3 |
| Mouse Liver | 1184×1184 | 558 | https://openorganelle.janelia.org/datasets/jrc_mus-liver |
| P7 Mouse Pancreas | 1184×1184 | 898 | https://openorganelle.janelia.org/datasets/jrc_mus-pancreas-4 |
| P7 Mouse Skin | 1184×1184 | 1231 | https://openorganelle.janelia.org/datasets/jrc_mus-skin-1 |
| FemFlyBrain | 1024×1024 | 1299 | https://neurodata.io/data/takemura13/ |
| FAFB | 1024×1024 | 700 | https://temca2data.org/ |
| Mouse Cortex | 1024×1024 | 700 | https://neurodata.io/data/kasthuri15/ |

## A1.2 Metrics

To quantitatively evaluate the performance of our method, we calculated several metrics, including Normalized Cross-Correlation (NCC), Mutual Information (MI), and Structural Similarity Index (SSIM) between the registrated image series and the ground truth (GT). These metrics provide a comprehensive evaluation of the accuracy of our alignment and registration results from different perspectives. Specifically, NCC measures the similarity between the result images and the GT, MI reflects the mutual information between the two, and SSIM evaluates the accuracy based on structural information. Dice is a set similarity metric commonly used to measure the similarity between two samples. Here, we use the Dice score to evaluate the segmentation accuracy of 3D segmentation results on the registered data.

## A2 Experimental Details

### A2.1 Dataset Setup

For training the trajectory tracking module, we split the original image data into 3000 image pairs $\{I_0, I_1\}$, where $I_0$ and $I_1$ represent adjacent images. These images are cropped to a size of $1184 \times 1184$. During training, the data is normalized to the range of [-1, 1]. The training-validation split is set to 0.95. Additionally, we randomly select one of the images from $\{I_0, I_1\}$ and apply random elastic deformation to simulate nonlinear distortions. Specifically, we first generate a random deformation field $D_{\mathrm{rand}}$, which indicates the pixel displacement matrix. This matrix is then smoothed using a Gaussian filter, and the resulting deformation is applied to create the deformed images. The deformation process is described by the following formulas:

$$\begin{cases} \phi_i = & \alpha \cdot \mathrm{size}(I_i) \cdot \mathrm{Gauss}(D_{\mathrm{rand}}, \sigma \cdot \mathrm{size}(I_i)), \\ I_i = & \phi_i \circ I_i, \end{cases} \tag{17}$$

where $\phi_i$ represents the random deformation field applied to image $I_i$, $\circ$ denotes the deformation operation, $D_{\mathrm{rand}}$ is the generated random displacement field, $\mathrm{Gauss}(\cdot)$ is the 2D Gaussian filter operator, $\alpha$ controls the displacement magnitude, and $\sigma$ determines the smoothing extent. In our experiments, we set $\sigma = 0.08$ and $\alpha = 1.0$.

During the testing phase, we cropped all six datasets into image stacks of size $1184 \times 1184 \times N$, where $N$ denotes the number of slices. Additionally, we applied random elastic deformations to all images except the first one to simulate nonlinear distortions. The deformation parameters were set to $\sigma = 0.08$ and $\alpha = 1.0$. Detailed information regarding the datasets can be found in Table 7. Our test data consists of several hundreds to around 1000 samples, which fully demonstrates our method's capability in handling long sequences and addressing the challenge of error accumulation.

## A2.2 Network Architecture and Training Details

The trajectory tracking network adopts a 2D U-Net architecture with residual multi-kernel fusion, consisting of 1) a cascaded feature encoder with hybrid convolutional blocks, and 2) a dense feature decoder with transposed upsampling. The encoder progressively reduces spatial resolution through four strided convolutions (stride=2), doubling the number of channels from an initial 8 to 64. Specifically, the encoder is composed of four convolutional layers with output channels of 8, 16, 32, and 64, respectively. Each convolutional layer includes two convolution operations: the first convolution uses a $3 \times 3$ kernel, followed by a second convolution with a larger $7 \times 7$ kernel to expand the receptive field, similar to the approach in LKUnet [28]. The decoder consists of four layers and reconstructs dense predictions using four transposed convolutions (kernel size = 2, stride = 2), halving the number of channels from 64 to 2. Skip connections concatenate the encoder features with the corresponding decoder layers to facilitate hierarchical feature fusion. The final displacement field is generated through dual $3 \times 3$ convolutions with a Softsign activation. To enable spatial warping, a Spatial Transformer Network (STN) [25] is employed for image registration. The model is optimized using the ADAM optimizer with parameters $\beta_1 = 0.5$ and $\beta_2 = 0.999$. The learning rate is set to $1 \times 10^{-4}$, and training is performed for 500 epochs with a batch size of 16. For the regularization term in the loss function, we set $\lambda = 1.5$ to encourage smoothness in displacement field.

Neural ODEs employ a 3D U-Net architecture to parameterize $f_\theta$. The number of network layers and the number of channels per layer are kept consistent with the 2D U-Net used in the trajectory tracking module, but large convolutional kernels are avoided to reduce memory consumption. For training, we use the ADAM optimizer with parameters $\beta_1 = 0.5$ and $\beta_2 = 0.999$. The learning rate is set to $1 \times 10^{-4}$, and training is conducted for 500 epochs with a batch size of 4. For the loss function, we set $\lambda = 0.1$ and $\mu = 0.005$. For the Gauss-Seidel method used in trajectory smoothing, we set the number of iterations to 200 and $\lambda = 0.1$. For Neural ODEs, we use the simple Euler solver, with all other parameters kept at their default settings.

## A2.3 Training and Testing on Real Data

A key advantage of our method lies in its unsupervised learning paradigm, which is particularly valuable for real-world datasets that lack reliable ground-truth correspondences. The training process on real data is divided into two stages: training of the trajectory tracking module and training of the trajectory smoothing module. We begin by cropping a small spatiotemporal block of size $H \times W \times T$ from the real dataset. Training samples are then constructed following the procedure described in Section A2.1, except that no additional elastic deformation is applied. The trajectory tracking network is subsequently trained in an unsupervised manner, identical to the approach used for synthetic data. For training the trajectory smoothing module, we first apply the previously trained network to obtain estimated trajectories. These are then used for the unsupervised training of the smoothing module, again following the same procedure as used for synthetic data.

For testing on real data, the networks trained on cropped data blocks can be directly applied to the full image stack. We adopt a sliding-window registration strategy to perform registration across the entire long image sequence. Specifically, given the full real image stack $\{I_i\}_{i=0}^{N-1}$, we define a receptive field of moderate size (about 100 slices). Within each windowed region, the trajectory tracking network is used to estimate the displacement trajectories. These trajectories are then refined by the trajectory smoothing module. Subsequently, inverse warping is applied based on the smoothed trajectories to perform image registration. The registered image block is then placed back into its corresponding location in the full stack. The window is shifted along the sequence, and the procedure is repeated until the entire image stack has been registered.

# A3 Supplementary Mathematical Derivations

## A3.1 Background of the Gauss-Seidel Method

The Gauss-Seidel method [18], a classical iterative solver for linear systems $A\mathbf{x} = \mathbf{b}$, that leverages the most recent updates to accelerate convergence, particularly in sparse, tridiagonal systems. This system can be equivalently written as a symmetric tridiagonal linear system:

$$A\mathbf{p} = \mathbf{b}, \tag{18}$$

where: - $\mathbf{p} = [\mathcal{P}(1), \ldots, \mathcal{P}(N-1)]^T$ is the unknown trajectory, - $\mathbf{b} = [\lambda \mathcal{P}^{(0)}(1), \ldots, \lambda \mathcal{P}^{(0)}(N-1)]^T$, - and $A \in \mathbb{R}^{(N-1) \times (N-1)}$ is a tridiagonal matrix with structure:

$$
A = \begin{bmatrix} 4+\lambda & -2 & & & \\ -2 & 4+\lambda & -2 & \ddots & \\ & \ddots & \ddots & \ddots & \\ & & -2 & 4+\lambda & -2 \\ & & & -2 & 4+\lambda \end{bmatrix}.
\tag{19}
$$

Applying the Gauss-Seidel method to this system, the general update rule for the $i$-th variable is:

$$
x_i^{(k+1)} = \frac{1}{A_{ii}} \left( b_i - \sum_{j<i} A_{ij} x_j^{(k+1)} - \sum_{j>i} A_{ij} x_j^{(k)} \right).
\tag{20}
$$

For our tridiagonal matrix, each row $z$ only has nonzero entries at $z-1$, $z$, and $z+1$. Substituting these into the general Gauss–Seidel formula, the update rule for $\mathcal{P}(z)$ becomes:

$$
\mathcal{P}^{(k+1)}(z) = \frac{1}{A_{zz}} \left( \lambda \mathcal{P}^{(0)}(z) - A_{z,z-1} \mathcal{P}^{(k+1)}(z-1) - A_{z,z+1} \mathcal{P}^{(k)}(z+1) \right),
\tag{21}
$$

where: - $A_{zz} = 4 + \lambda$, - $A_{z,z-1} = A_{z,z+1} = -2$.

Substituting these in yields:

$$
\mathcal{P}^{(k+1)}(z) = \frac{1}{4+\lambda} \left( 2\mathcal{P}^{(k+1)}(z-1) + 2\mathcal{P}^{(k)}(z+1) + \lambda \mathcal{P}^{(0)}(z) \right).
\tag{22}
$$

For simplicity, we divide both numerator and denominator by 2, resulting in the final explicit Gauss-Seidel update:

$$
\mathcal{P}^{(k+1)}(z) = \frac{\mathcal{P}^{(k+1)}(z-1) + \mathcal{P}^{(k)}(z+1) + \frac{\lambda}{2} \mathcal{P}^{(0)}(z)}{2+\lambda}, \quad z = 1, \ldots, N-1.
\tag{23}
$$

### A3.2 Derivation of the Gauss-Seidel Iteration Update Formula

For our trajectory smoothing method, the Gauss-Seidel iteration can be adapted by introducing a fidelity constraint. Below is the detailed derivation of the iteration formula. Specifically, to minimize the energy function:

$$
E(u) = \lambda \sum_{i=0}^{N-1} (u_i - u_i^{(0)})^2 + \sum_{i=1}^{N-2} (u_{i-1} - 2u_i + u_{i+1})^2,
\tag{24}
$$

we take the partial derivative of $E$ with respect to each $u_i$ and set it to zero to obtain the optimality condition.

The fidelity term contributes a derivative of:

$$
\frac{\partial}{\partial u_i} \left( \lambda (u_i - u_i^{(0)})^2 \right) = 2\lambda (u_i - u_i^{(0)}).
\tag{25}
$$

The smoothing term consists of a sum of squared second-order differences, where each point $u_i$ appears in multiple overlapping terms, together with $u_{i-1}$ and $u_{i+1}$. For clarity, we focus on a representative term where $u_i$ is the center, to illustrate how the smoothness constraint acts locally on the trajectory:

$$
\frac{\partial}{\partial u_i} (u_{i-1} - 2u_i + u_{i+1})^2 = -4(u_{i-1} - 2u_i + u_{i+1}).
\tag{26}
$$

Combining the derivatives of both terms, we obtain the total gradient:

$$
\frac{\partial E}{\partial u_i} = 2\lambda (u_i - u_i^{(0)}) - 4(u_{i-1} - 2u_i + u_{i+1}) = 0.
\tag{27}
$$

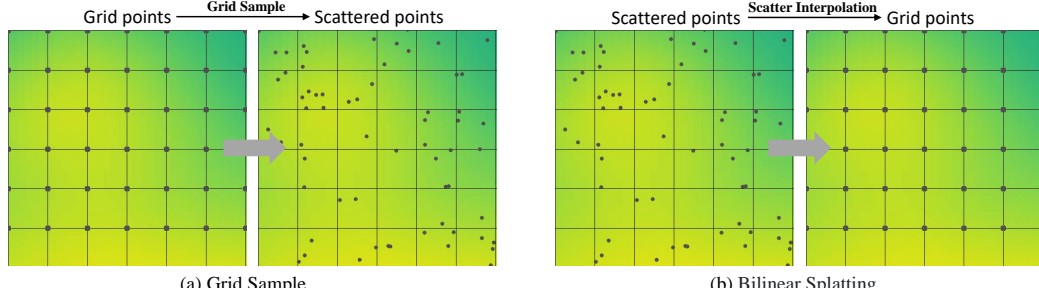

(a) Grid Sample             (b) Bilinear Splatting

Figure 6: Regular grid sampling and Bilinear splatting. Grid sampling is used to obtain trajectory coordinates at sub-pixel resolution. For trajectory inversion, bilinear splatting is used to estimate the displacement at pixel grid locations.

Expanding and rearranging terms gives:

$$2\lambda(u_i - u_i^{(0)}) - 4u_{i-1} + 8u_i - 4u_{i+1} = 0. \tag{28}$$

Solving for $u_i$, we derive the following linear equation:

$$(2\lambda + 8)u_i = 2\lambda u_i^{(0)} + 4u_{i-1} + 4u_{i+1}. \tag{29}$$

This leads to the Gauss-Seidel iteration update formula:

$$u_i^{(k+1)} = \frac{1}{2\lambda + 8}\left(2\lambda u_i^{(0)} + 4u_{i-1}^{(k+1)} + 4u_{i+1}^{(k)}\right), \tag{30}$$

Simplifying further, we obtain

$$u_i^{(k+1)} = \frac{1}{\lambda + 2}\left(\lambda u_i^{(0)} + u_{i-1}^{(k+1)} + u_{i+1}^{(k)}\right), \tag{31}$$

where $\lambda$ is halved from its original value. We use the updated $u_{i-1}^{(k+1)}$ from the current iteration and the old $u_{i+1}^{(k)}$ from the previous iteration to update $u_i^{(k+1)}$.

## A4    Algorithmic Details of Bilinear Splatting

The goal of bilinear splatting is to map discrete feature points onto a regular grid using bilinear interpolation. Given a set of normalized coordinates and their associated feature vectors, we need to compute the grid values using the surrounding grid points. The coordinates are normalized to the range $[0, H)$ and $[0, W)$, where $H$ and $W$ are the height and width of the target grid. The feature vectors are associated with these coordinates and are projected to the grid using bilinear interpolation, as shown in Figure 6.

Let the coordinates of the discrete points be represented as coords $= \{(y_1, x_1), \ldots, (y_N, x_N)\}$, where each $(y_i, x_i)$ is a 2D coordinate within the normalized range. The corresponding feature vectors for each point are denoted by values $= \{v_1, \ldots, v_N\}$, where $v_i \in \mathbb{R}^C$ is the feature vector for point $i$, and $C$ is the feature dimension.

To perform bilinear splatting, we begin by identifying the four nearest grid points that surround each continuous coordinate $(y_i, x_i)$. These grid points form the corners of the smallest axis-aligned square in the discrete grid that encloses the given coordinate. Specifically, we obtain them by applying the floor and ceiling operations to both the $y$- and $x$-coordinates:

$$\begin{cases} y_{\text{floor}} = \lfloor y_i \rfloor, & x_{\text{floor}} = \lfloor x_i \rfloor, \\ y_{\text{ceil}} = \lfloor y_i \rfloor + 1, & x_{\text{ceil}} = \lfloor x_i \rfloor + 1. \end{cases} \tag{32}$$

These points form the basis for distributing the value at $(y_i, x_i)$ across the surrounding pixels based on their bilinear interpolation weights.

Next, the weights for bilinear splatting are calculated based on the fractional parts of the coordinates. Let $\Delta y = y_i - \lfloor y_i \rfloor$ and $\Delta x = x_i - \lfloor x_i \rfloor$ represent the fractional parts of the coordinates. The four interpolation weights are given by:

$$\begin{cases} w_1 = (1 - \Delta x)(1 - \Delta y), & w_2 = (1 - \Delta x)\Delta y, \\ w_3 = \Delta x(1 - \Delta y), & w_4 = \Delta x \Delta y. \end{cases} \tag{33}$$

These weights represent the contribution of each neighboring grid point to the interpolated value at the target coordinate $(y_i, x_i)$. Now, we distribute the feature values to the corresponding grid locations based on these weights. For each coordinate $(y_i, x_i)$, we update the feature grid by adding the weighted value of the point to the grid locations corresponding to the four neighboring points. This is done using the following scatter operation:

$$\begin{cases} \text{Grid}[y_{\text{floor}}, x_{\text{floor}}] = \text{Grid}[y_{\text{floor}}, x_{\text{floor}}] + v_i \cdot w_1, \\ \text{Grid}[y_{\text{ceil}}, x_{\text{floor}}] = \text{Grid}[y_{\text{ceil}}, x_{\text{floor}}] + v_i \cdot w_2, \\ \text{Grid}[y_{\text{floor}}, x_{\text{ceil}}] = \text{Grid}[y_{\text{floor}}, x_{\text{ceil}}] + v_i \cdot w_3, \\ \text{Grid}[y_{\text{ceil}}, x_{\text{ceil}}] = \text{Grid}[y_{\text{ceil}}, x_{\text{ceil}}] + v_i \cdot w_4. \end{cases} \tag{34}$$

The grid values are accumulated at the corresponding grid positions, and the sum of the weights at each grid location is also accumulated for normalization purposes.

Finally, to ensure proper averaging when multiple points contribute to the same grid position, we normalize the resulting grid by the sum of the weights:

$$\text{Grid}[y, x] = \frac{\sum v_i \cdot w_i}{\sum w_i} \tag{35}$$

To avoid division by zero in case of no contribution to a grid location, a small epsilon $\epsilon = 1e - 8$ is added to the denominator during the normalization step. Thus, the bilinear splatting operation efficiently interpolates the feature values from discrete points to the target grid, with smooth interpolation properties ensured by the bilinear weights.

## A5  Limitations and Future Work

Our approach can be divided into three main modules: 1) pixel trajectory tracking, 2) dynamics-based trajectory optimization, and 3) trajectory inversion and image registration. For each module, we conducted a simple exploration in the ablation study (see Sec.B.1) to evaluate its effect on registration performance, including the influence of complex scenarios such as high-noise artifacts and wrinkles that may affect trajectory-tracking accuracy. Future work may investigate 3D registration methods that are robust to artifacts and wrinkles, as well as explore the use of more advanced architectures architectures for improving registration accuracy and generalizability. Furthermore, Table 11 presents the registration accuracy and execution time on the Mus Liver-3 dataset, where the test slice resolution is set to $1024 \times 1024$. Our method achieves the highest SSIM, demonstrating superior accuracy compared to other approaches. However, due to the computational demands of the ODE solver, it does not achieve the fastest runtime.

To further investigate runtime efficiency, we conducted additional experiments focusing on two optimization strategies: (i) sparse trajectory sampling (every 2/3/4 pixels); and (ii) resolution reduction by downsampling followed by upsampling. Table 9 shows that sparse sampling substantially reduces runtime (from 2.46s to 0.95s) while maintaining state-of-the-art accuracy, with SSIM only slightly decreasing from 0.874 to 0.831. Table 10 demonstrates that reducing the resolution also improves efficiency (from 2.46s to 0.91s), though at the cost of accuracy (SSIM drops from 0.874 to 0.827). Nevertheless, the performance remains superior to the GaussReg baseline. We attribute the observed degradation primarily to the use of a simple bilinear upsampling strategy. We argue that could achieve a more favorable balance between runtime and accuracy.

Table 8: Execution time and performance.

|  | TrakEM2 [56] | EFSR [76] | EMReg [42] | SEMLeSS [53] | GaussReg [82] | Ours |
|---|---|---|---|---|---|---|
| Time(s) | 3.92 | 3.44 | 0.28 | 27.11 | **0.92** | 2.46 |
| SSIM | 0.66 | 0.52 | 0.65 | 0.61 | 0.82 | **0.87** |

Table 9: Ablation study of sparse sampling.

|  | All pixels | 1/2 pixel | 1/3 pixel | 1/4 pixel |
|---|---|---|---|---|
| SSIM | 0.874 | 0.866 | 0.842 | 0.831 |
| Time(s) | 2.46 | 1.35 | 1.12 | 0.95 |

Table 10: Ablation study of resolution reduction.

|  | Full resolution | 1/2 resolution | 1/4 resolution |
|---|---|---|---|
| SSIM | 0.874 | 0.845 | 0.827 |
| Time(s) | 2.46 | 1.12 | 0.91 |

## A6 Broader Impacts and Discussion

Our work focuses on 3D registration for Series Section Electron Microscopy (ssEM). One of the key advantages of our approach is its fully unsupervised training, which eliminates the reliance on ground truth and makes it applicable to real-world ssEM scenarios. For ssEM applications [79, 13, 15], this is particularly important for the increasingly complex and diverse electron microscopy image datasets [15, 77], as acquiring comprehensive ground truth across various tissue and cell types is challenging. Our method can be applied to the modeling and analysis of biological cell tissues, assisting researchers in exploring the structure and function of cellular tissues.

For other downstream tasks in ssEM, such as 3D segmentation [40, 41, 47], our method provides well-structured image data that aligns with natural biological morphology. By improving the axial continuity of the raw image data while preserving the structural integrity of superstructural details, our approach can enhance the performance of downstream tasks.

For the design of 3D registration algorithms, we propose a novel paradigm from the perspective of trajectory optimization. Our approach can be divided into three main components: 1) pixel trajectory tracking, 2) dynamics-based trajectory optimization, and 3) trajectory inversion and image registration. These three modules are decoupled and interchangeable. In theory, each module can be replaced with a more efficient algorithm. For example, more efficient feature point tracking algorithms [11, 10] or optical flow estimation algorithms [62, 78] can be used for trajectory tracking. Low-pass filtering [26] and convex quadratic programming (QP) [12, 17] can be used for trajectory smoothing. Radial basis function interpolation [73] and other splatting techniques [3] can be used for trajectory inversion. This helps researchers explore more efficient 3D registration methods based on this paradigm.

Due to time and resource constraints, we were unable to validate the performance of our method on larger-scale scenarios, particularly on ultra-high-resolution image data. The challenges posed by computational complexity in larger environments and the variability of data noise remain. In future work, we will refine our approach and adapt it into a microscopy image processing tool suitable for large-scale real-world datasets.

Table 11: Performance of different registration methods on downstream segmentation tasks across six datasets. We evaluate segmentation accuracy using the Dice score (Avg %).

|  | EMReg [42] | EFSR [76] | SEMLeSS [53] | TrakEM2 [56] | GaussReg [82] | Ours |
|---|---|---|---|---|---|---|
| Mus Heart | $0.35 \pm 0.06$ | $0.82 \pm 0.02$ | $0.65 \pm 0.05$ | $0.68 \pm 0.03$ | $\mathbf{0.89} \pm 0.01$ | $\underline{0.88} \pm 0.01$ |
| Mus Kidney | $0.41 \pm 0.07$ | $0.75 \pm 0.02$ | $0.67 \pm 0.02$ | $0.75 \pm 0.02$ | $\underline{0.82} \pm 0.01$ | $\mathbf{0.84} \pm 0.01$ |
| Mus Liver-3 | $0.41 \pm 0.08$ | $0.90 \pm 0.02$ | $0.86 \pm 0.02$ | $0.90 \pm 0.02$ | $\mathbf{0.95} \pm 0.01$ | $\underline{0.94} \pm 0.01$ |
| Mus Liver | $0.57 \pm 0.09$ | $0.89 \pm 0.04$ | $0.86 \pm 0.05$ | $0.92 \pm 0.04$ | $\mathbf{0.94} \pm 0.03$ | $\mathbf{0.94} \pm 0.03$ |
| Mus Pancreas | $0.46 \pm 0.04$ | $0.87 \pm 0.02$ | $0.79 \pm 0.02$ | $0.90 \pm 0.01$ | $\mathbf{0.94} \pm 0.01$ | $\underline{0.93} \pm 0.01$ |
| Mus Skin | $0.50 \pm 0.07$ | $0.87 \pm 0.03$ | $0.76 \pm 0.04$ | $0.90 \pm 0.02$ | $\mathbf{0.94} \pm 0.01$ | $\mathbf{0.94} \pm 0.03$ |

Table 12: Evaluation of the diffeomorphic property of deformation fields using the Folds metric (% of $|J\varphi| \leq 0$).

| % of $|J\phi| \leq 0$ | TrakEM2 [56] | EFSR [76] | EMReg [42] | SEMLeSS [53] | GaussReg [82] | Ours |
|---|---|---|---|---|---|---|
| Mus Heart | / | 0.188 | 0.243 | 0.135 | 0.099 | 0.0171 |
| Mus Kidney | / | 0.206 | 0.379 | 0.24 | 0.174 | 0.0098 |
| Mus Liver | / | 0.193 | 0.31 | 0.155 | 0.106 | 0.0124 |
| Mus Skin | / | 0.171 | 0.252 | 0.102 | 0.091 | 0.0095 |

# B  Additional Quantitative Results and Visualization

## B.1  More Experimental Results

**Robustness to Error Accumulation.**  In practical applications, serial section electron microscopy (ssEM) datasets often contain hundreds or even thousands of images. This large number of slices poses a challenge for long-sequence registration, as cumulative errors can easily arise, eventually leading to substantial sequence drift and compromising the accurate reconstruction of the biological specimen's true 3D structure. To systematically evaluate the ability of our method to suppress such cumulative errors, we conducted comparative experiments on six long-sequence datasets listed in Table 7, focusing on the robustness of our method versus GaussReg [82] and SEMLeSS [53].

Specifically, Figures 12 illustrate how the registration accuracy changes as the number of slices increases. The results show that our method consistently achieves higher average registration accuracy across the entire sequence, with smaller fluctuations in the accuracy curve. This indicates superior stability and robustness when handling long sequences. Moreover, Figures 7 and 8 present the 3D reconstruction results on the remaining four datasets. It is evident that our method accurately recovers the spatial structures of various biological tissues, further demonstrating its generalizability and robustness across different types of data.

**Performance on 3D Segmentation Tasks.**  3D segmentation [40, 41] is a important task in serial section electron microscopy (ssEM) and has been widely applied in various biological image analysis domains [19, 43]. High-quality 3D registration plays a crucial role in segmentation tasks, as it can significantly mitigate artifacts caused by structural deformation, scale variations, and differences in imaging modalities, thereby improving segmentation accuracy. To comprehensively evaluate the applicability of our method in 3D segmentation, we conducted experiments on six datasets involving various organelles, including nuclei and mitochondria. Specifically, we employed a segmentation network [67] to perform segmentation on the images registered by each method. The segmentation performance was quantified by computing the Dice similarity coefficient between the predicted results and the ground-truth (GT) label stacks.

Table 11 summarizes the Dice scores for 3D segmentation results across the six datasets. As shown, our method achieves the best performance on three datasets and the second-best on the remaining three, with overall results slightly below those of the supervised method GaussReg (with differences no greater than 0.01). This discrepancy may be attributed to the segmentation model's sensitivity to subtle misalignments, while our method focuses on global trajectory optimization and may have limitations in handling local details. For instance, as illustrated in Figure 8 for the Mus Liver-3 dataset, minor local misalignments can still be observed in our registration results. Nevertheless, our method attains accuracy comparable to that of the supervised GaussReg. Furthermore, Figs. 9, 10, and 11 present 3D visualizations of the segmentation results on multiple datasets. These visual results

Table 13: Sensitivity analysis of the hyperparameter $\lambda$.

| $\lambda$(SSIM) | 0.15 | 1.5 | 4.5 | 7.5 | 10 |
|---|---|---|---|---|---|
| Mus Heart | 0.855 | 0.872 | 0.87 | 0.864 | 0.83 |
| Mus Kidney | 0.827 | 0.851 | 0.855 | 0.843 | 0.809 |

Table 14: Impact of different network architectures on trajectory tracking performance.

| SSIM | RAFT [64] | GAFlow [45] | SEA-RAFT [71] |
|---|---|---|---|
| Mus Heart | 0.892 | 0.917 | 0.909 |
| Mus Kidney | 0.894 | 0.933 | 0.925 |
| Mus Liver | 0.871 | 0.928 | 0.918 |
| Mus Skin | 0.92 | 0.943 | 0.932 |

Table 15: Effect of large deformations on registration performance.

| | GaussReg ($\alpha$=1.0) | Ours ($\alpha$=1.0) | GaussReg ($\alpha$=1.5) | Ours ($\alpha$=1.5) | GaussReg ($\alpha$=2.0) | Ours ($\alpha$=2.0) |
|---|---|---|---|---|---|---|
| Mus Heart | 0.832 | 0.872 | 0.811 | 0.867 | 0.772 | 0.856 |
| Mus Kidney | 0.781 | 0.851 | 0.763 | 0.843 | 0.739 | 0.823 |
| Mus Liver | 0.805 | 0.842 | 0.782 | 0.836 | 0.758 | 0.817 |
| Mus Skin | 0.827 | 0.864 | 0.797 | 0.851 | 0.763 | 0.838 |

Table 16: Effect of noise levels on registration performance.

| | GaussReg (5%) | Orgin (5%) | GaussReg (10%) | Orgin (10%) | GaussReg (15%) | Orgin (15%) |
|---|---|---|---|---|---|---|
| Mus Heart | 0.815 | 0.851 | 0.788 | 0.832 | 0.748 | 0.771 |
| Mus Kidney | 0.762 | 0.833 | 0.734 | 0.822 | 0.715 | 0.759 |
| Mus Liver | 0.779 | 0.814 | 0.74 | 0.784 | 0.72 | 0.714 |
| Mus Skin | 0.804 | 0.837 | 0.751 | 0.803 | 0.728 | 0.742 |

demonstrate that our method can reliably reconstruct the correct 3D structure of biological tissues, leading to clearer and more consistent segmentation boundaries, further validating its practicality in downstream tasks.

In addition, we report the Folds metric in Table 12 to further evaluate the diffeomorphic property of the deformation fields. The results demonstrate that our method consistently achieves the lowest Folds values, remaining around 0.01, whereas most baseline methods report values greater than 0.1. This indicates that our approach substantially reduces grid folding in the generated deformation fields, thereby better preserving topological consistency compared to existing baselines.

**Results on Real-World Data.**    To further evaluate the performance of our method on real-world datasets, we present the registration results on three datasets, FemFlyBrain, FAFB[3], and the Mouse Cortical Dataset, as shown in Figures 13-17. The results demonstrate that our method achieves accurate and consistent alignment across diverse biological samples, effectively handling complex morphological variations and imaging artifacts.

**More Ablation Experiments**    To gain a deeper understanding of our method, we conducted additional ablation studies. Table 13 presents a sensitivity analysis of the key hyperparameter $\lambda$, which primarily controls the smoothness term in the loss function of the trajectory tracking module. Table 14 investigates the impact of adopting more advanced network architectures on the performance of the trajectory tracking module. As shown, replacing the baseline with a more sophisticated network slightly improves the registration accuracy, demonstrating the modular flexibility of our framework and its compatibility with advanced architectures. Tables 15 and 16 further compare registration performance under more challenging conditions, including large deformations and high noise levels. The results indicate a gradual performance degradation as deformation or noise intensity increases, suggesting promising directions for extending our approach to more complex real-world scenarios.

---

[3]Due to FAFB data acquisition, only our registration results are shown.

## B.2 Explainable Visualization Study

To better understand the performance and explainability of our new paradigm, we conducted a visualization analysis of the intermediate trajectories. Specifically, we performed trajectory tracking on data with nonlinear distortions, ground truth data, and registration results, ensuring consistency between the trajectories across different datas. The visualization results are shown in Figure 18. It is evident from the figure that the trajectories of the nonlinear distorted data exhibit irregular noise jitter, leading to abrupt changes in local curvature, which reflect the impact of distortion on the motion trajectory. In contrast, the trajectory of the ground truth data maintains the natural evolutionary pattern of the organism's movement, exhibiting smooth and continuous changes. Our method successfully overcame abnormal physical deformations in the registration results, faithfully and precisely restoring the natural motion trajectory.

## B.3 Failure Cases

We observe that our method is to some extent dependent on the accuracy of the trajectory tracking module. When the tracking is suboptimal, the subsequent image registration performance may be affected. This issue becomes particularly prominent when handling real-world anisotropic data, where the axial resolution is often significantly lower than the lateral (XY) resolution. Substantial structural and textural differences arise between adjacent slices (as shown in Figure 19). Moreover, real datasets often suffer from high noise levels, imaging artifacts, and missing slices, further complicating reliable trajectory estimation. In Figure 20, we illustrate several representative failure cases, in which the axial resolution of the image stacks is approximately one-tenth of the lateral resolution, with noticeable noise and missing slices. Although our method is still capable of performing registration under such challenging conditions, the visual quality may degrade due to the aforementioned interfering factors. We hope future research will explore more robust and efficient trajectory tracking modules, as well as 3D registration methods that are better suited for anisotropic and noisy real-world data.

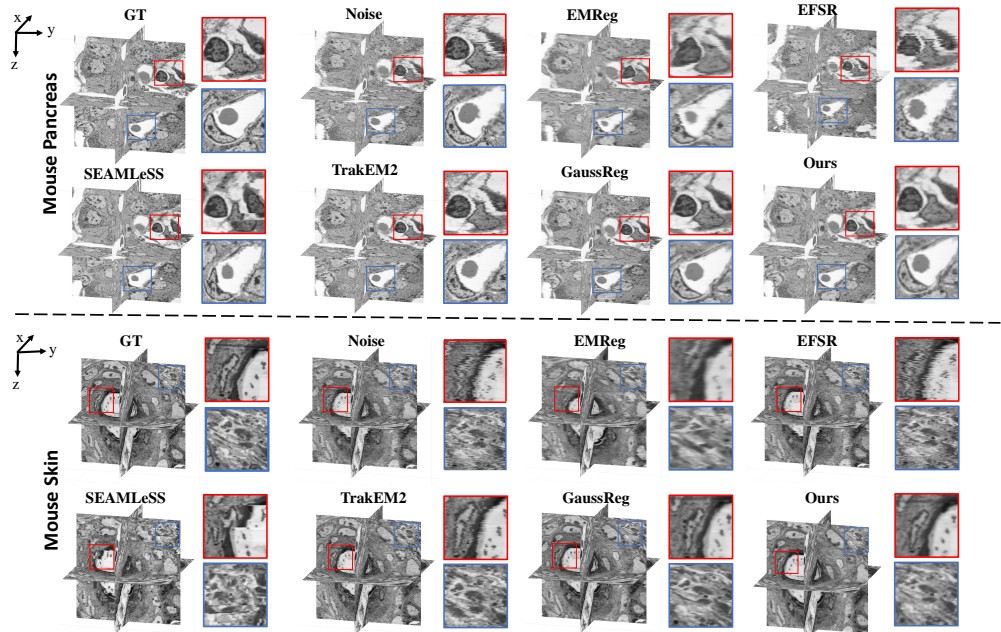

Figure 7: More 3D visualization of registration results on Mus Pancreas and Mus Skin datasets.

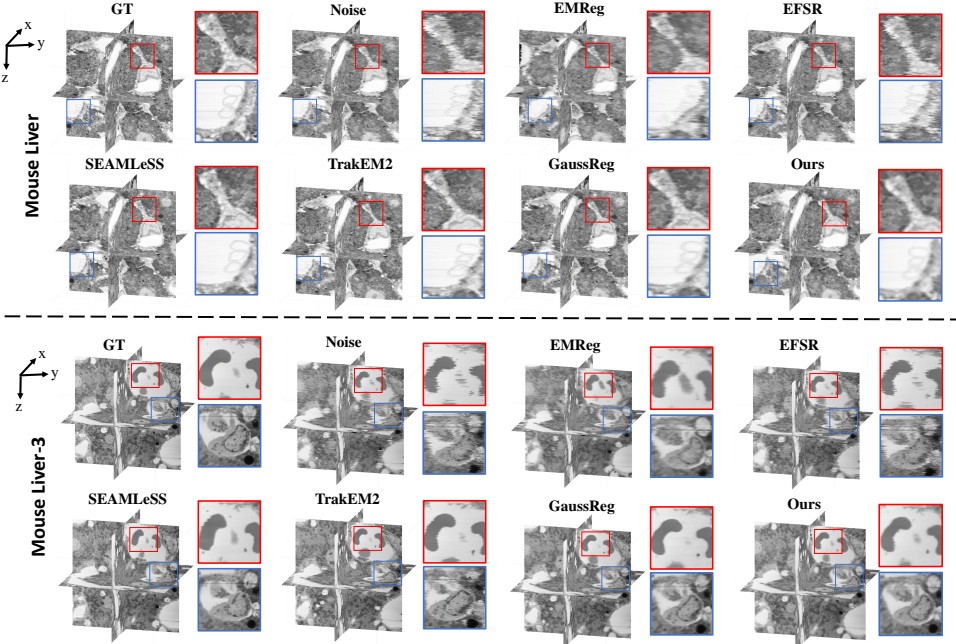

Figure 8: More 3D visualization of registration results on Mus Liver and Mus Liver-3 datasets.

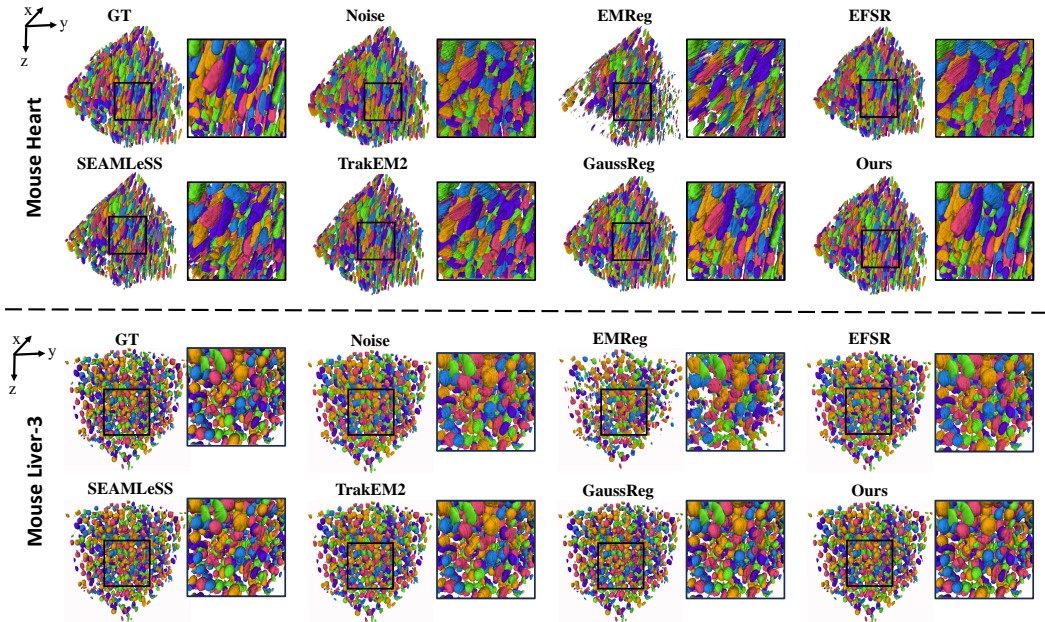

Figure 9: 3D segmentation visualizations of registration results using various methods on the Mus Heart and Mus Liver-3 datasets.

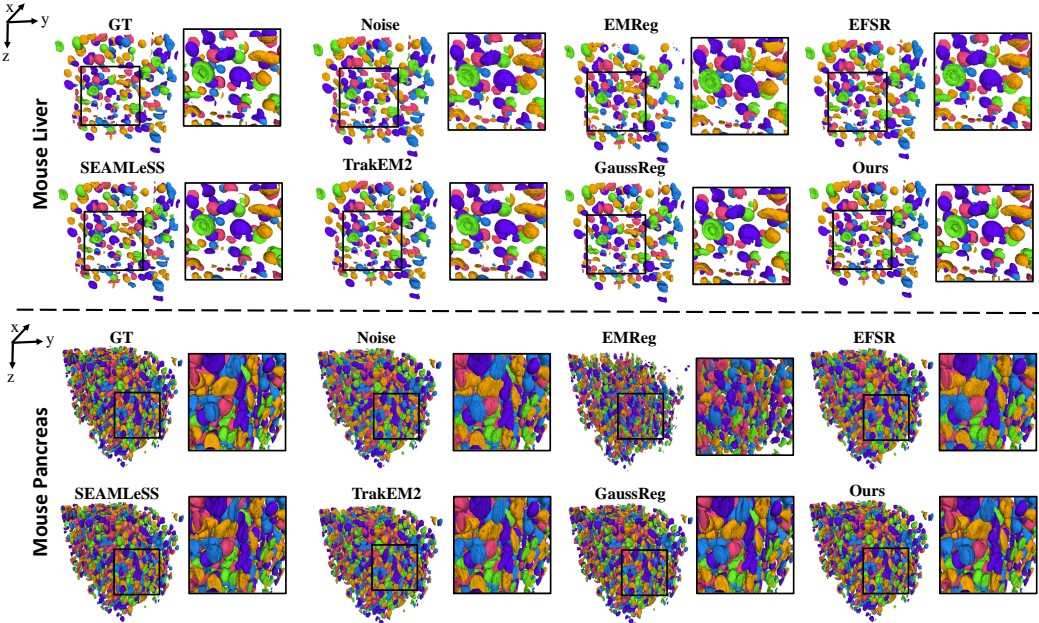

Figure 10: 3D segmentation visualizations of registration results using various methods on the Mus Liver and Mus Pancreas datasets.

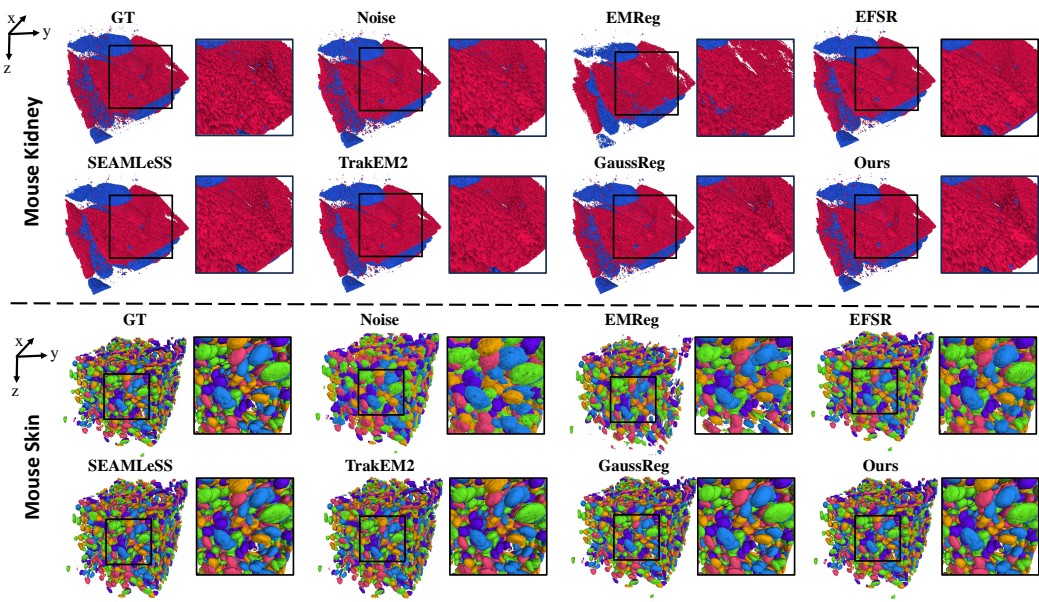

Figure 11: 3D segmentation visualizations of registration results using various methods on the Mus Kidney and Mus Skin datasets.

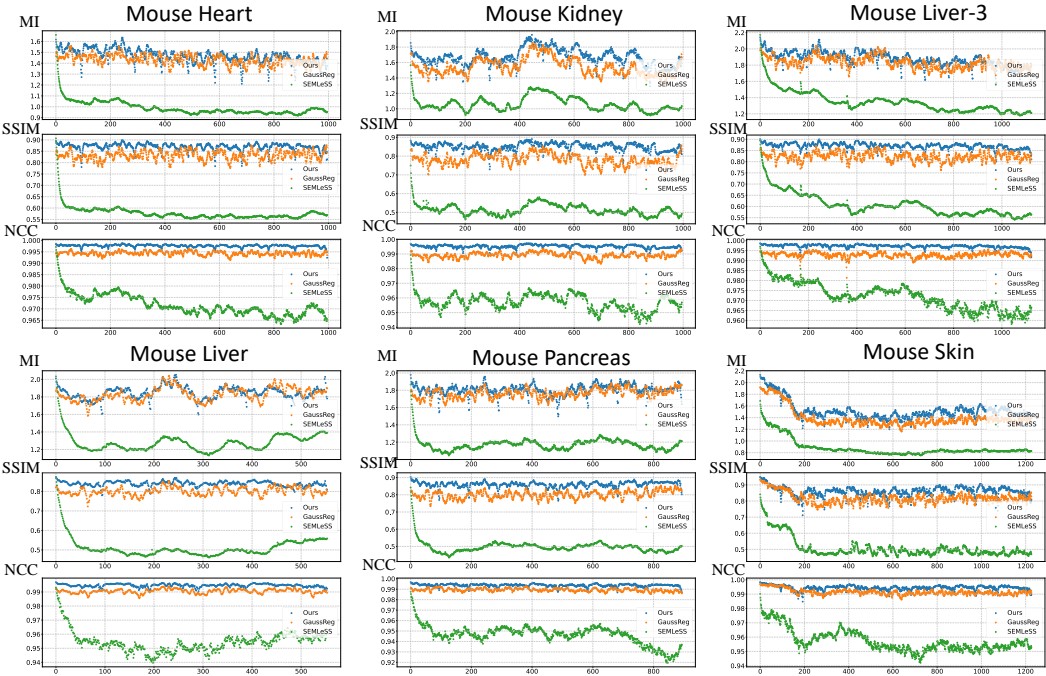

Figure 12: Error accumulation comparison between GaussReg [82] and SEMLeSS [53] and ours on six dataset.

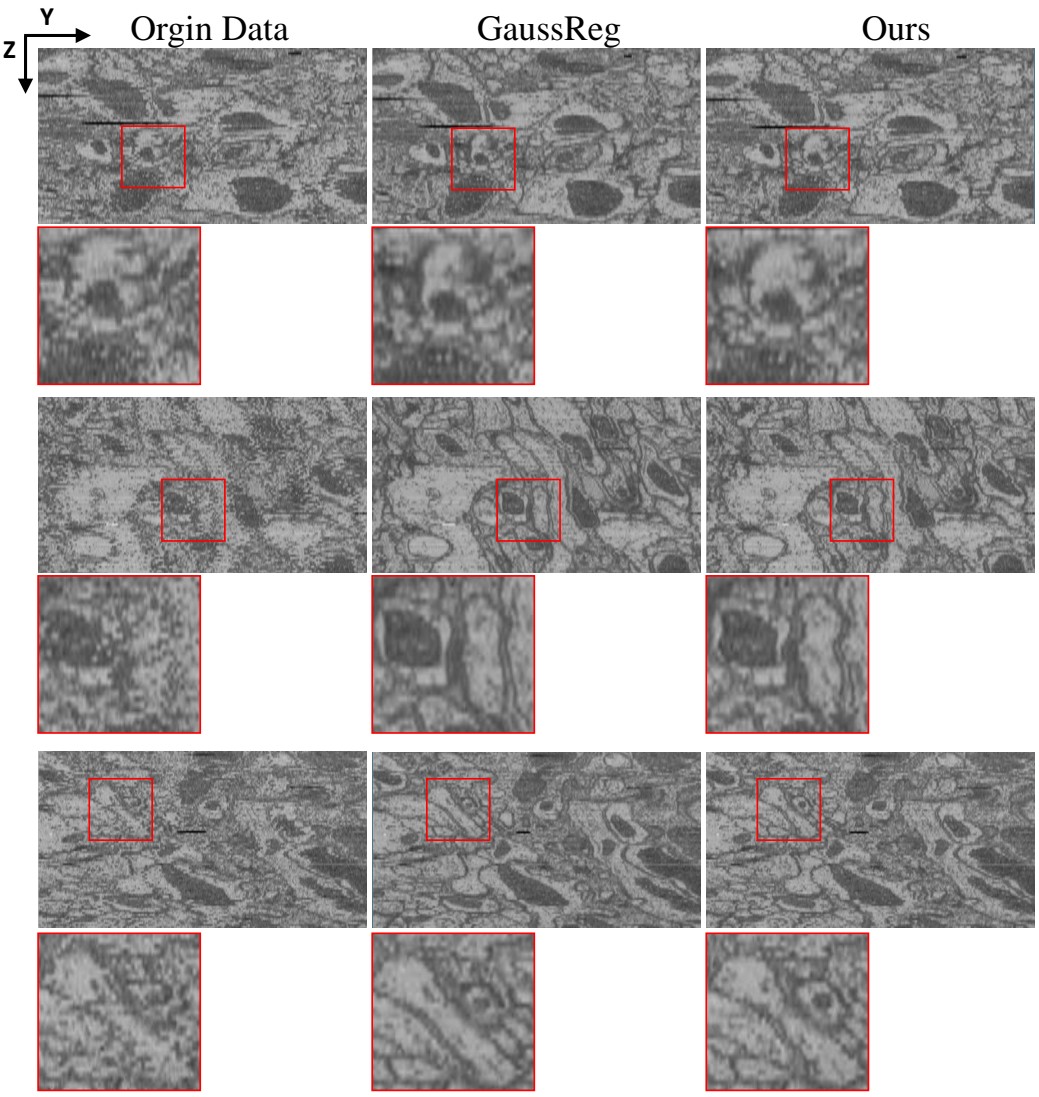

Figure 13: Side views of the original data, the registration results of GaussReg [82], and our method on the FemFlyBrain dataset [63].

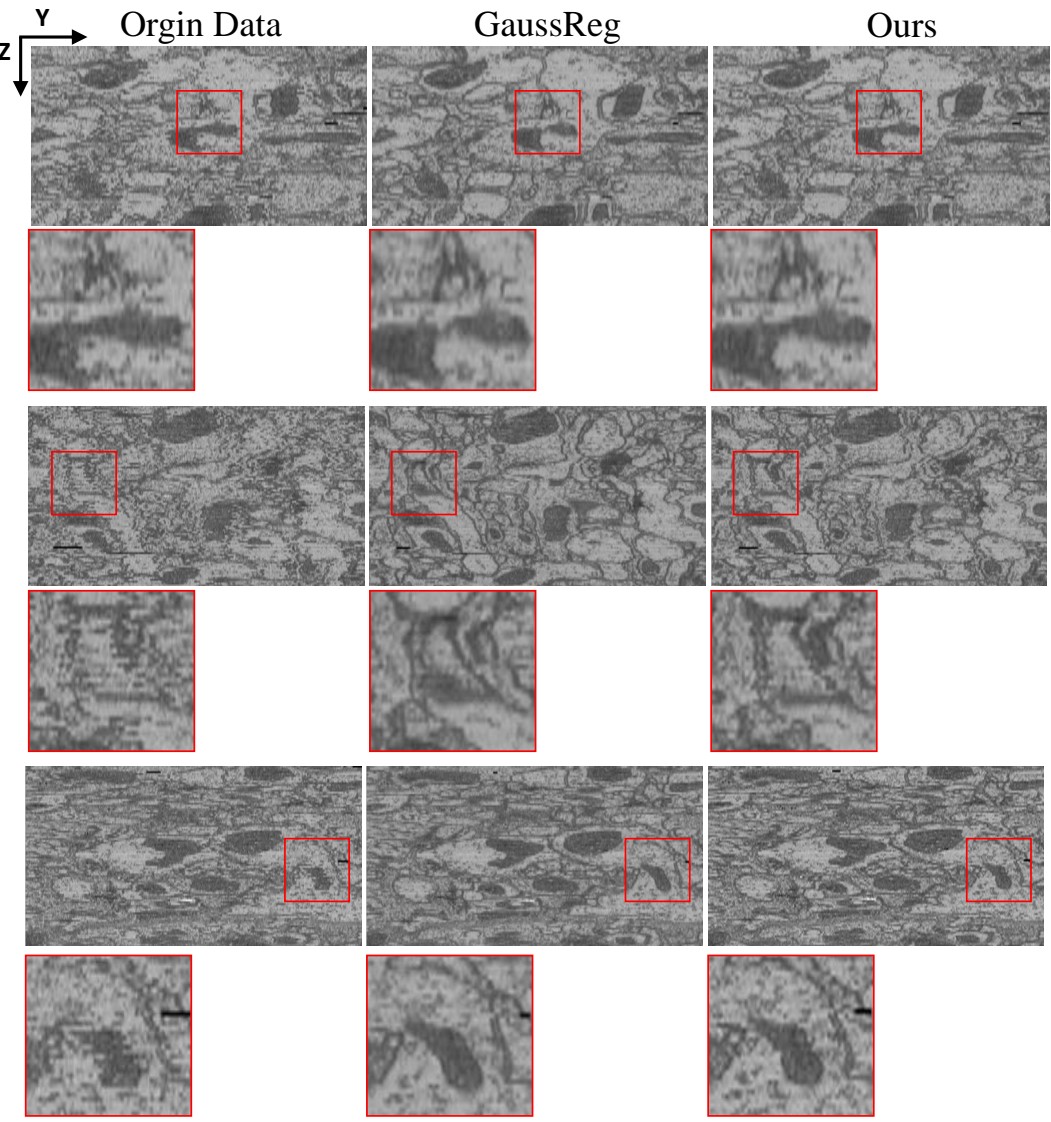

Figure 14: Side views of the original data, the registration results of GaussReg [82], and our method on the FemFlyBrain dataset [63].

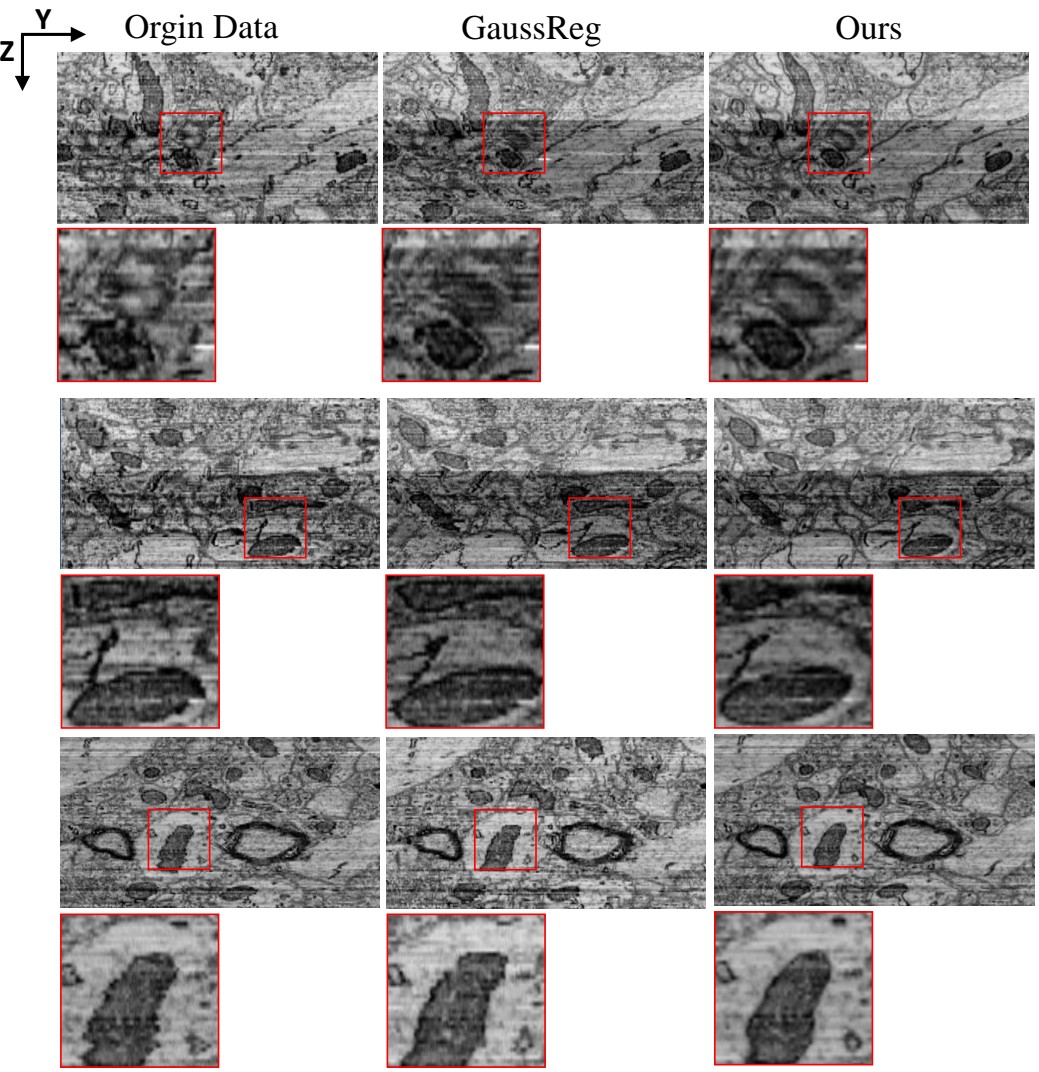

Figure 15: Side views of the original data, the registration results of GaussReg [82], and our method on the Mouse cortical dataset [32].

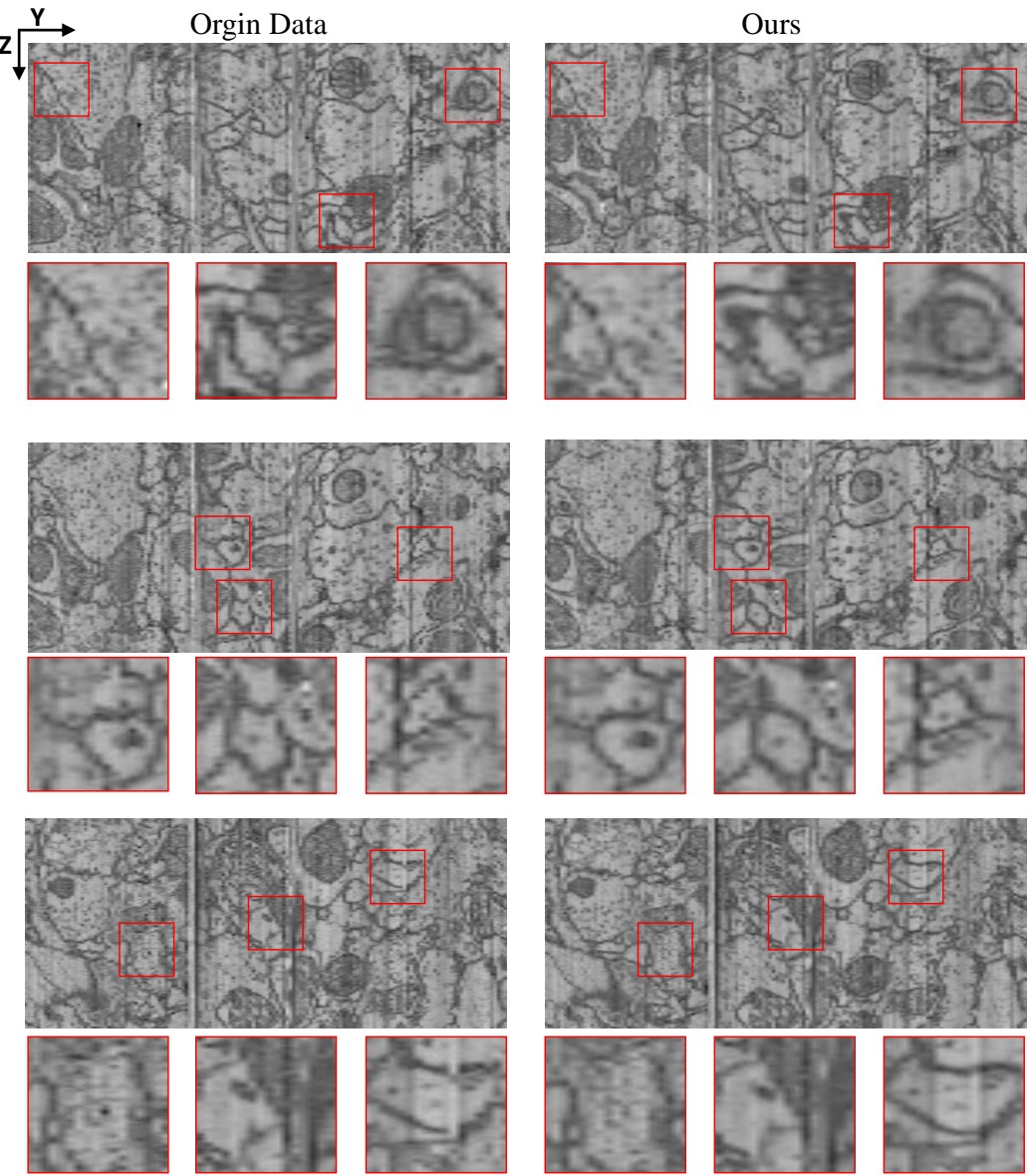

Figure 16: Side views of the original data, the registration results on the FAFB dataset [84].

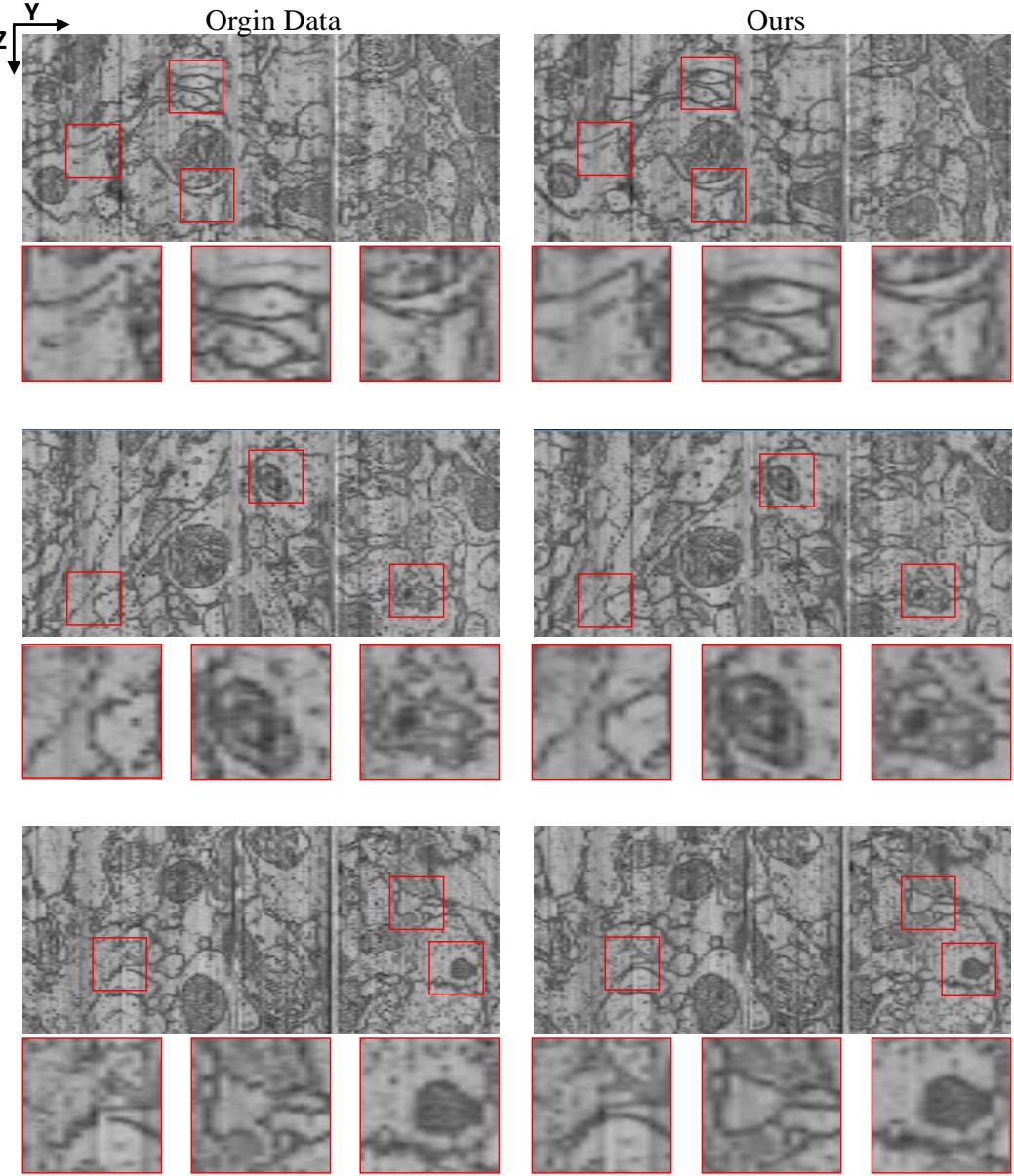

Figure 17: Side views of the original data and the registration results on the FAFB dataset [84].

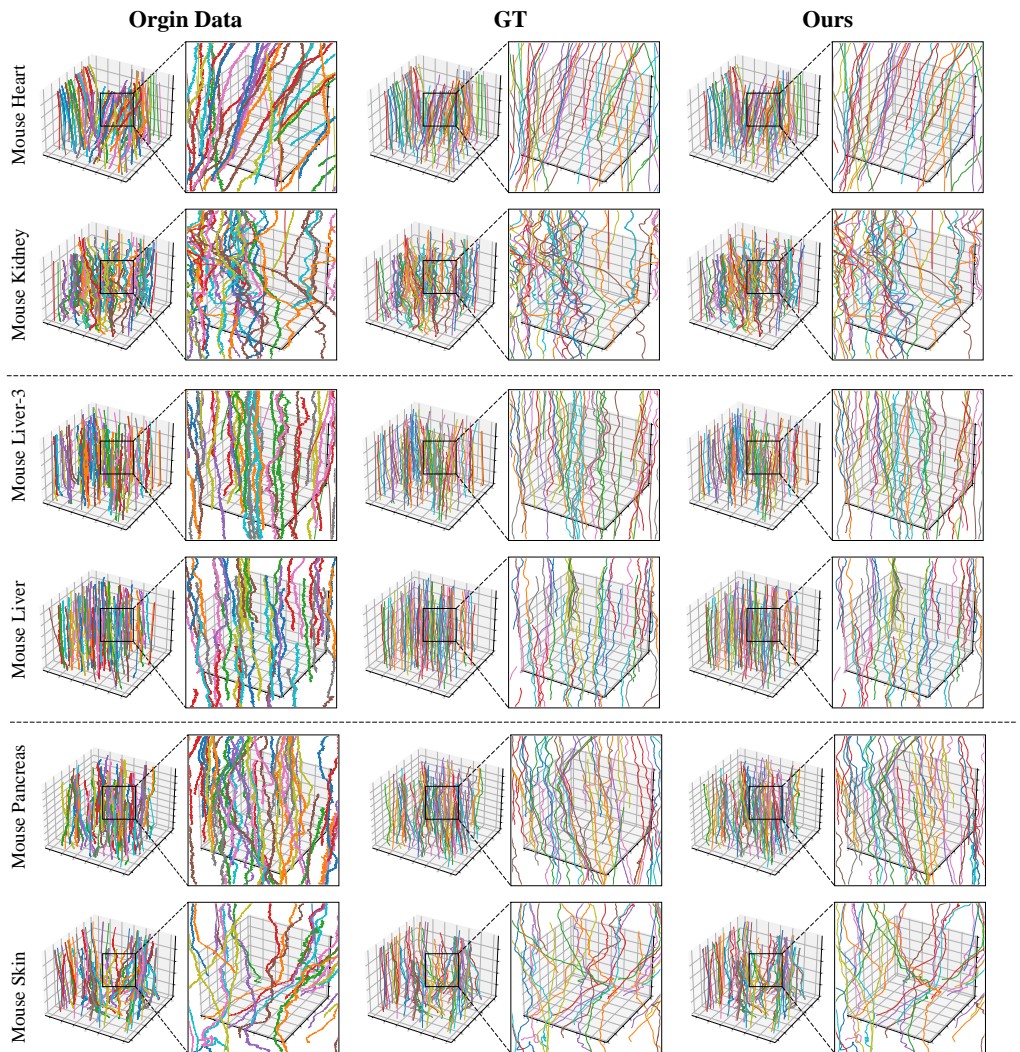

Figure 18: Trajectory visualization of the original data, ground truth, and registration results.

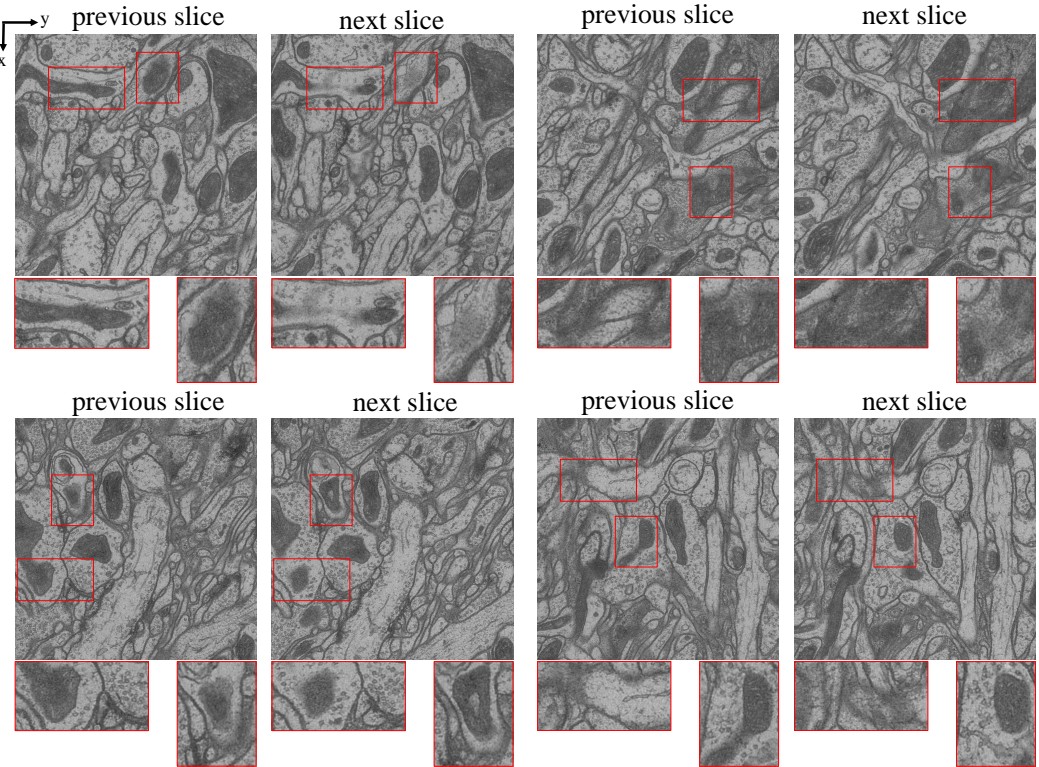

Figure 19: Four examples showing texture structure differences between adjacent slices of anisotropic data.

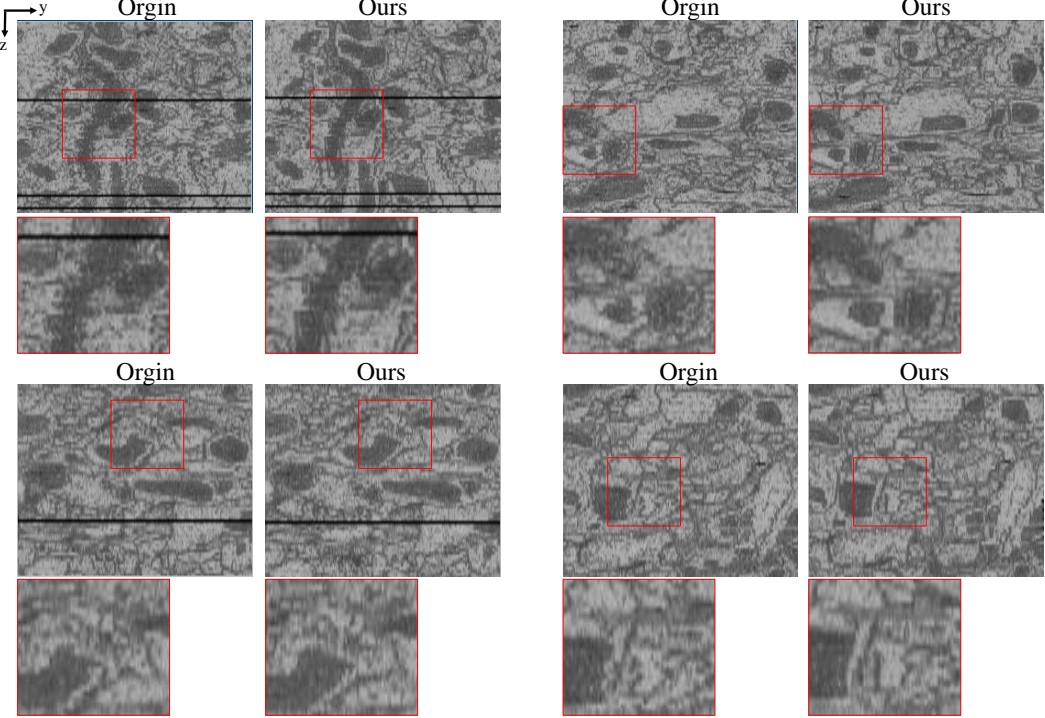

Figure 20: Four failure cases from the FemFlyBrain dataset. These examples are affected by low axial resolution, missing slices, and high noise.

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
