# OpenReview forum: "Unsupervised Trajectory Optimization for 3D Registration in Serial Section Electron Microscopy using Neural ODEs"
_NeurIPS.cc/2025/Conference — NeurIPS 2025 poster_

### Official Review · Reviewer_TWfx · 2025-06-18

**Clarity:** 3
**Significance:** 3
**Originality:** 3
**Rating:** 5
**Confidence:** 4

**Summary:**

This paper addresses the challenge of 3D registration for Series Section Electron Microscopy (ssEM), a critical step for reconstructing biological structures that is often hampered by nonlinear distortions and the accumulation of errors during physical sectioning. The paper proposes a fundamental shift, moving from traditional local, pairwise registration to a global optimization framework. They redefine 3D registration as a continuous manifold trajectory optimization problem. This formulation is motivated by the physical insight that biological tissue deforms smoothly, whereas sectioning artifacts introduce abrupt, non-physical curvature mutations into the motion paths of cellular structures as they move through the image stack.
The method consists of three main stages:
1. Pixel Trajectory Tracking: An initial, noisy trajectory for each pixel is established by using a 2D U-Net to estimate displacement fields between adjacent slices.
2. Dynamics-based Trajectory Optimization: In the core contribution, a hybrid solver smooths the noisy trajectories. It combines the Gauss-Seidel method, a classical, iterative solver for rapid, stable convergence based on a physical model with a Neural Ordinary Differential Equation, which serves as a data-driven compensator to learn and correct for complex, nonlinear deviations. An adaptive weighting scheme dynamically balances these two components.
3. Trajectory Inversion and Registration: The final, smoothed trajectories are used to compute a displacement field, which is mapped back to the discrete pixel grid via an efficient bilinear splatting technique to produce the final deformation fields.
A key advantage of the pipeline is its fully unsupervised training, making it highly applicable to real-world ssEM data where ground-truth alignments are rarely available.

**Questions:**

1. The ablation study justifies using the Euler solver for the Neural ODE to balance performance and cost. To better understand this trade-off, could you please:

   * Quantify the concrete runtime differences between the Euler, Dopri5, and RK4 solvers on your test hardware?
   * Clarify if you used the adjoint sensitivity method for more efficient gradient computation during training? If not, have you considered it or other continuous-depth models to improve the speed/accuracy profile, given that runtime is a key limitation?

      2. The framework's modularity is presented as a strength, suggesting that more efficient algorithms could be swapped in for each component. Given the pipeline's acknowledged dependence on the initial tracking module, could you clarify the practicalities of this modularity? For example, if the 2D U-Net were replaced by a state-of-the-art sparse feature tracker like TAPIR, would the trajectory smoothing module require complete retraining to adapt to the different statistical properties of sparse vs. dense trajectories? How does the module generalize?

      3. Could you provide any theoretical or empirical analysis regarding the convergence of the combined Gauss-Seidel + Neural ODE solver, particularly in relation to the behavior of α?

      4. The loss hyperparameters (λ, μ) show a clear performance peak. How were the search ranges determined, and how well do you expect these values to generalize to new datasets without another extensive search?

      5. Have you tested on synthetic data with known artifacts (folds, missing sections) beyond those shown in failure cases?

**Ethical Concerns:**

["NO or VERY MINOR ethics concerns only"]

**Final Justification:**

I appreciate the authors’ detailed rebuttal. The authors have addressed key weaknesses, such as adding more data validation and improving runtime trade-offs.

**Limitations:**

Yes.

**Paper Formatting Concerns:**

No concerns.

**Quality:**

3

**Strengths And Weaknesses:**

Strengths:
* The paper's primary strength is its conceptual reframing of registration as a global trajectory optimization problem. This is an elegant and significant contribution that directly addresses the fundamental issue of error accumulation that plagues traditional pairwise methods.
* The fully unsupervised nature of the framework is a major practical advantage, making it applicable to diverse ssEM datasets where obtaining ground-truth annotations is prohibitive.
* The methodology is of high quality, featuring a sophisticated hybrid solver that combines a physics-based prior with data-driven learning in a theory-guided, data-corrected approach.
* The experimental validation is extensive and robust. The method is tested across six diverse tissue types and compared against multiple state-of-the-art baselines.
* The authors provide derivations, algorithmic details, and a thorough discussion of limitations and failure cases in the supplementary material, which greatly enhances the paper's quality and reproducibility.

Weaknesses:
* A significant weakness is the method's computational expense, which the authors acknowledge is due to the ODE solver. While presented as a trade-off for accuracy, this could be a barrier to adoption for labs needing to process the very large-scale datasets that motivate the work. The reporting is also incomplete, with detailed runtime analysis missing for most datasets.
* The entire pipeline's performance is critically dependent on the quality of the initial trajectory tracking module. This module's architectural design uses a 2D U-Net that operates on strictly local, adjacent slice pairs, giving it no wider temporal context. This lack of context makes the tracker prone to errors in regions of low texture or ambiguity. As acknowledged by the authors and demonstrated in the paper's failure cases, these initial tracking errors can propagate through the pipeline, leading to suboptimal registration results that the global smoothing module cannot fully correct. This fundamentally restricts the method's robustness, especially when applied to challenging real-world data with significant noise or anisotropy.
* While the formulation is innovative, the paper lacks a theoretical convergence analysis for the novel combined Gauss-Seidel + Neural ODE framework. The conditions under which the adaptive blending mechanism (Eq. 11) is guaranteed to converge are not discussed.
* There is limited analysis of key parameters like β in the adaptive weighting. While Table 4 shows ablation for Gauss-Seidel iterations (50-300) and Table 3 provides hyperparameter grid search for λ and μ, the sensitivity analysis for β is missing and the hyperparameter studies were conducted only on the EFPL dataset, not across all datasets.
* The method's quantitative evaluation relies on synthetic distortions created by applying a Gaussian-smoothed random displacement field. This model may not fully represent the diverse range of real-world artifacts from physical sectioning, such as compressive shearing, chatter, or large tissue folds. Consequently, the method's strong performance on this specific type of simulated noise may not guarantee generalization to all distortion types encountered in practice.
* In the final stage, continuous trajectories are resampled onto a discrete grid by bilinear splatting which introduces an unevaluated risk to the preservation of fine biological structures. This interpolation method inherently smooths information as it distributes a single point's value across multiple pixels, which could be problematic to the single-voxel precision required for connectomics. Hence, the potential for this operator to blur these high-frequency details is a significant concern.

---

> ### Author Rebuttal · Authors · 2025-07-31
>
> #### **Weaknesses 1**
> Thank you for your suggestion. We have explored the speed–accuracy trade-off for large-scale EM data. Our optimization significantly reduces runtime (from 2.46s to 0.95s), while preserving SOTA SSIM accuracy (see our responses to Cons 3 from Reviewer L5DT).
>
>
> #### **Weaknesses 2**
> We agree that trajectory tracking plays an important role in the overall pipeline performance. Our current design adopts a lightweight 2D U-Net to enable fast and memory-efficient tracking between adjacent slices, but we acknowledge that its limited temporal context may cause errors in ambiguous regions. To address this, we explored two strategies to improve performance under more challenging scenarios (see our responses to Q.2 from Reviewer Luv5).  But we acknowledge the importance of modeling longer temporal context, and we plan to explore trajectory models with extended time horizons to handle challenging scenarios with low texture or strong noise.
>
> Moreover, to evaluate robustness, we tested our method on two additional real datasets (see responses to Cons 1 from Reviewer L5DT). All corresponding results will be included in the revision.
>
> #### **Weaknesses 3**
> We apologize for the typo in Eq. (10) of the main text (implementation and experiments were correct). The original equation:
> $$
> \mathcal{P}^{(t_N)} = \alpha \cdot \mathcal{G}(\mathcal{P}^{(t_0)}) + (1-\alpha) \cdot \mathcal{F}_\theta(\mathcal{P}^{(t_0)}),
> $$
>
> where $\mathcal{F}_\theta(\mathcal{P}^{(t_0)})$ denotes a shorthand for $\text{ODESolver}$
>
> should be corrected to:
>
> $$
> \mathcal{P}^{(t_N)} = (1-\alpha) \cdot \mathcal{G}(\mathcal{P}^{(t_0)}) + \alpha \cdot \mathcal{F}_\theta(\mathcal{P}^{(t_0)}).
> $$
>
> Define
> $$
> \mathcal{P}^* := \mathcal{G}(\mathcal{P}^{(t_0)}), \quad \mathcal{P}^\dagger := \mathcal{F}_\theta(\mathcal{P}^{(t_0)}).
> $$
>
> 1. **Gauss-Seidel:**
> Converges under standard conditions for SPD linear systems, solving the quadratic objective
> $$
> \mathcal{L}(\mathcal{P}) = \sum_{z=1}^{N-1} \|\mathcal{P}(z-1) - 2\mathcal{P}(z) + \mathcal{P}(z+1)\|^2 + \lambda \sum_{z=1}^{N-1} \|\mathcal{P}(z) - \mathcal{P}^{(0)}(z)\|^2,
> $$
> via iterations
> $$
> \mathcal{P}^{(k+1)} = T \mathcal{P}^{(k)} + c,
> $$
> with spectral radius $\rho(T) < 1$.
>
> 2. **Neural ODE:**
> Then the ODE has a unique solution $\mathcal{P}(t)$ on $[t_0, t_N]$, and numerical solvers yield approximations with bounded error:
> $$
> \|\mathcal{P}_{\text{num}}^{(t_N)} - \mathcal{P}^{(t_N)}\| \leq \epsilon.
> $$
>
> 3. **Adaptive Blending:**
> - Both modules converge individually.
> - Blending weight $\alpha$ increases monotonically as error decreases via a sigmoid function.
> - If $\|\mathcal{P}^* - \mathcal{P}^\dagger\|$ is sufficiently small, the combined update is stable and converges.
>
> Under these conditions, the adaptive blending update $\mathcal{P}^{(t_N)}$ converges reliably.
>
>
> #### **Weaknesses 4**
> We thank the reviewer for emphasizing the importance of parameter \$\beta\$ in Eq. (11). It controls the sigmoid steepness and thus the transition speed between Gauss-Seidel and Neural ODE modules. A larger \$\beta\$ makes \$\alpha\$ shift faster from 0 to 1, enabling a quicker switch, while a smaller \$\beta\$ yields a smoother transition. We will add convergence curves and performance comparisons for different \$\beta\$ values in the revised version.
>
> Ablation studies were also repeated on four additional datasets (see responses to Weaknesses 2 from Reviewer zfxm).
>
> #### **Weaknesses 5**
> We employ Gaussian-smoothed random displacement fields to ensure fair comparison. Specifically, we strictly follow the deformation protocols established in widely used baselines such as GaussReg, EFSR, and EMReg, which introduce slice-wise perturbations that closely mimic real-world misalignments observed in ssEM data.
>
> We agree with the reviewer that this model may not fully represent the diverse range of real-world artifacts resulting from physical sectioning. Therefore, in addition to the four real FemFlyBrain blocks included in the supplementary material, we further conduct experiments on six blocks from two widely used real datasets (see our responses to Cons 1 from Reviewer L5DT), providing stronger validation under real-world scenarios. Additional results will be included in the revision.
>
>
> #### **Weaknesses 6**
> Thank you for the thoughtful question. Our trajectory inversion step aims to obtain displacement vectors defined on a pixel-aligned grid, which are then used to construct the full deformation field.
>
> It is important to note that this inversion operates on displacements rather than pixel intensities. But it does not distribute a single point’s value across multiple pixels, and thus does not introduce the kind of smoothing effect typically associated with image-based interpolation. The resulting deformation field is applied once to warp the input stack using STN [1]. Therefore, bilinear splatting is only used as a technical means to discretize continuous displacements, and it does not additionally degrade spatial precision.
>
> [1] Spatial transformer networks. NeurIPS 2015
>
> #### **Q.1**
> Thank you for the suggestion. We have added runtime comparisons of Euler, Dopri5, and RK4 solvers on datasets beyond EFPL (see Table 13). The results show that while Dopri5 and RK4 provide marginal improvements in performance compared to Euler (e.g., SSIM increases from 0.872 to 0.877 and 0.876 on the mouse heart dataset), they come at a significant computational cost (runtime increases from 2.45s to 8.33s and 7.68s, respectively). Given this trade-off, the Euler solver offers a better balance between accuracy and efficiency in our pipeline. For the training of Neural ODEs, we already employ the adjoint sensitivity method to enable memory-efficient and scalable gradient computation.
>
> **Table13**
> |Dataset|Solver|MusHeartTime(s)|MusHeartSSIM|MusKidneyTime(s)|MusKidneySSIM|MusLiverTime(s)|MusLiverSSIM|MusSkinTime(s)|MusSkinSSIM|
> |-|-|-|-|-|-|-|-|-|-|
> ||Euler|2.45|0.872|2.41|0.851|2.50|0.842|2.46|0.864|
> ||Dopri5|8.33|0.877|8.52|0.855|8.71|0.848|8.39|0.869|
> ||RK4|7.68|0.876|7.66|0.854|7.72|0.847|7.25|0.868|
>
> #### **Q.2**
> Thank you for your suggestion. We explored replacing the original U-Net with more advanced architectures while keeping the same training pipeline (see our responses to Cons 2 from Reviewer L5DT). The results show that incorporating more advanced models can further improve performance, yielding 0.3–0.5 higher SSIM scores.
>
> Different dense prediction modules don’t require retraining the trajectory smoothing module. However, sparse prediction are unsuitable because our trajectory inversion uses bilinear splatting to get displacement vectors on the pixel grid. Sparse trajectories can cause missing pixel values and reduce registration quality. Since accurate registration needs a dense, continuous deformation field, dense prediction modules are preferable.
>
> We have added ablation studies on trajectory sampling intervals (see our responses to Cons 3 from Reviewer L5DT). The results demonstrate that overly sparse trajectory sampling leads to performance degradation (e.g., SSIM drops from 0.874 to 0.831), especially for EM images where subpixel-level alignment is critical.
>
>
> #### **Q.3**
> Thank you for your insightful question. We have provided a theoretical analysis regarding the convergence of the combined Gauss-Seidel + Neural ODE solver in our above response to Weaknesses.3. As for the behavior of the weighting coefficient $\alpha$, it is dynamically modulated by the relative residual through a sigmoid-shaped function controlled by the parameter $\beta$. Specifically, a larger $\beta$ causes $\alpha$ to transition more rapidly from 0 to 1 once the relative error has sufficiently decreased, thereby accelerating the switch from the linear Gauss-Seidel module to the nonlinear Neural ODE module. In contrast, a smaller $\beta$ leads to a more gradual and smoother transition (see our above response to Weaknesses 4).
>
> #### **Q.4**
> We thank the reviewer for raising this important question. The $(\lambda, \mu)$ search ranges were initially determined through a grid search on a small subset of the EFPL dataset during preliminary experiments. To assess generalization, we further conducted ablation studies on additional datasets (see responses to Weaknesses 2 from Reviewer zfxm) and consistently observed clear performance peaks around the same parameter values. These results suggest that our choice of $(\lambda, \mu)$ offers good generalization across datasets, reducing the need for extensive re-tuning when applying our method to new domains.
>
> #### **Q.5**
> We thank the reviewer for the suggestion. We have added new results on four datasets with varied inter-slice intervals to simulate missing sections (see Table 14). Due to time constraints, we were unable to retrain the trajectory tracking module, resulting in reduced performance as the slice interval increases. However, based on our ablation studies on the tracking module (see our response to Cons 2 from Reviewer L5DT), we believe that training on anisotropic data and replacing the tracker with more advanced networks can effectively mitigate this issue.
>
> As for folded sections, handling them is particularly challenging because it requires not only detecting the folds but also unfolding the distorted tissue, which is a complex nonlinear operation [2]. In practice, it is common to treat folded slices as missing sections and exclude them from the registration process to avoid introducing errors [3]. We will include these clarifications and relevant discussions in the revision.
>
> [2] A unified deep learning framework for ssTEM image restoration. IEEE TMI, 2022.
>
> [3] A complete electron microscopy volume of the brain of adult Drosophila melanogaster. Cell, 2018.
>
> **Table14**
> |SSIM\Intervals|1|2|4|6|
> |-|-|-|-|-|
> |MusHeart|0.872|0.851|0.832|0.804|
> |MusKidney|0.851|0.832|0.811|0.796|
> |MusLiver|0.842|0.828|0.801|0.795|
> |MusSkin|0.864|0.850|0.835|0.807|

---

> ### Comment · Reviewer_TWfx · 2025-08-03
>
> I appreciate the authors’ detailed rebuttal. The authors have addressed key weaknesses such as adding more data validation and improving runtime trade-offs, but shared concerns remain about deployment practicality, scalability, and clarity of writing.

---

> > ### Comment · Area_Chair_Ao4U · 2025-08-03
> >
> > Dear Min,
> >
> > Many thanks for reading the rebuttal and deciding to retain your score. I think that's perfectly reasonable! I'd be grateful however if you could add 1-2 sentences that ultimately justify this choice in the context of the original text and the rebuttal text.
> >
> > best wishes, AC

---

> > ### Author Response · Authors · 2025-08-04
> >
> > Dear Reviewer TWfx,
> >
> > Thank you sincerely for your valuable time, your positive assessment of our work, and your thoughtful response to our rebuttal. We are happy to clarify your concerns regarding **deployment practicality**, **scalability**, and **clarity of writing**.
> >
> > **Deployment Practicality**
> >
> > First, we have systematically evaluated the trade-off between speed and accuracy. As mentioned in our responses to your Weaknesses 1 and Reviewer L5DT's Cons 3, our optimizations reduce the runtime from **2.46s** to **0.95s**, while maintaining SOTA-level SSIM accuracy. Here, we provide detailed data and result analysis.
> >
> > Specifically, we adopted two efficiency strategies:
> >
> > - (i) **Sparse trajectory sampling** ( see Table 15);
> > - (ii) **Resolution reduction** via downsampling followed by upsampling (see Table 16).
> >
> > Our experiments show that sparse sampling reduces the runtime to **0.95s**, which is close to the fastest baseline GaussReg at **0.92s**, while achieving an SSIM of **0.831**, outperforming GaussReg’s **0.82**. Resolution reduction further improves the speed to **0.91s** with an SSIM of **0.827**, which still surpasses GaussReg. Since we currently use a simple bilinear upsampling strategy, we believe that combining these two strategies can achieve a better balance between efficiency and accuracy.
> >
> > Table15
> > |Sampling Strategy|SSIM|Time(s)|
> > |-|-|-|
> > |All pixels|0.874|2.46|
> > |Every 2nd pixel|0.866|1.35|
> > |Every 3rd pixel|0.842|1.12|
> > |Every 4th pixel|0.831|0.95|
> >
> > Table16
> > |Resolution|SSIM|Time(s)|
> > |-|-|-|
> > |Full resolution|0.874|2.46|
> > |1/2 resolution|0.845|1.12|
> > |1/4 resolution|0.827|0.91|
> >
> > Second, our framework can support **parallel block-wise processing** and scales to TB-scale ssEM volumes (see our response to Reviewer zfxm's Q4):
> >
> > 1. Divide the volume into blocks
> > 2. Run trajectory tracking and inversion in parallel on multiple GPUs
> > 3. Stitch local deformation fields
> > 4. Apply the global deformation
> >
> > For datasets with thousands of slices, we split them into blocks of 100 slices, allowing processing on GPUs with as little as **40GB** or even less memory (see A2.3 of Supplementary Materials).
> >
> > **Scalability**
> >
> > We have also explored improving the performance by replacing the trajectory tracker with stronger optical flow networks (see our responses to your Q2 and Reviewer L5DT's Cons 2). As shown in Table 17, using more advanced networks leads to SSIM improvements of **0.3--0.5**. Despite the simplicity of our current U-Net architecture for ease of demonstrating the advantage of our new framework for the ssEM 3D registration, it already outperforms GaussReg, and allows for future enhancement through more powerful tracking modules.
> >
> > Table17
> > |SSIM|RAFT(ECCV2020)|GAFlow(ICCV2023)|SEARAFT(ECCV2024)|
> > |-|-|-|-|
> > |MusHeart|0.892|0.917|0.909|
> > |MusKidney|0.894|0.933|0.925|
> > |MusLiver|0.871|0.928|0.918|
> > |MusSkin|0.920|0.943|0.932|
> >
> > We fully understand your emphasis on **deployment** and **scalability**, as these are important directions for future research. We believe that such extensions are most meaningful after establishing a fundamentally new and effective paradigm for ssEM 3D registration. Our work represents exactly such a **paradigm shift**, rather than incremental improvements over prior methods. However, we will follow your suggestion to add sufficient discussions on deployment and scalability as well in our revision.
> >
> > **Clarity of Writing.**
> >
> > We have corrected the formula errors, figure misinterpretations, and citation mistakes pointed out in your Weaknesses 3 and in Reviewer zfxm’s Weaknesses 1 and Q2. If there still remain any clarity issues, we would be happy to revise further to improve the quality of this work.
> >
> > Once again, we truly appreciate your thoughtful attention to the deployment practicality, scalability, and clarity of writing. We hope that our responses address your concerns and you could let us know if there is anything else we can further clarify.

---

### Official Review · Reviewer_zfxm · 2025-07-02

**Clarity:** 2
**Significance:** 2
**Originality:** 3
**Rating:** 5
**Confidence:** 4

**Summary:**

This paper proposes a novel unsupervised framework for 3D registration in ssEM by formulating it as a continuous trajectory optimization problem. Combining Gauss-Seidel iteration with Neural ODEs, the method balances global smoothness with local structure preservation without requiring ground truth. Experiments across diverse tissue types show improved accuracy and robustness over existing approaches.

**Questions:**

1. The benchmark dataset generation should be further clarified: Please clarify how the synthetic benchmarks are constructed, particularly with respect to the OpenOrganelle datasets. Are the original volumes considered as ground truth, with artificial z-axis distortions added to simulate noise? If so, what procedure is used to introduce these deformations, and how is the realism of such synthetic noise validated?


2 The paper benchmarks against models such as EMReg and TrakEM2. However, EMReg, while primarily pairwise, can be adapted to optimize over larger blocks with global regularization. This appears to contradict the claim that prior methods are strictly local. Could the authors clarify how EMReg and other baselines were implemented or adapted for the comparison, particularly in terms of their global vs. local optimization capabilities? Also, note that the EMReg citation in Table 1 (ref [29]) is inconsistent with its mention in the main text (ref [24]); please ensure citation accuracy.


3. The comparison with biophysically-informed alternatives can be made. A key contribution of this work is the integration of biophysical priors into the registration process. However, other methods such as SOFIMA (https://github.com/google-research/sofima)(based on finite element modeling) similarly aim to preserve biological morphology through elastic deformation modeling. It would strengthen the paper to compare against such methods that also explicitly model biomechanical plausibility. If models like this are out of scope for direct benchmarking, a discussion of the similarities and differences in modeling philosophy would still be helpful.


4. The paper would benefit from additional detail on how the method scales to TB-scale ssEM datasets. Specifically, how is memory usage managed for pixel-wise trajectory tracking across thousands of slices, and are any tiling, chunking, or multi-CPU strategies employed? Can users apply the model with a 40 GB memory GPU? Including this information would clarify the practical feasibility of the method for real-world datasets.

**Ethical Concerns:**

["NO or VERY MINOR ethics concerns only"]

**Final Justification:**

The authors have addressed my concerns in the response.

**Limitations:**

1. The proposed method optimizes global pixel trajectories, which may introduce higher computational overhead compared to locally constrained models. Given the large size of typical EM volumes, runtime is a critical factor in practical deployment. It would be valuable for the authors to report time consumption and discuss the trade-off between registration accuracy and computational cost relative to existing methods.

2. Many real-world ssEM pipelines include stitching as a pre-alignment step. Some baseline methods such as TrakEM2 are capable of both stitching and registration. It would strengthen the paper to clarify whether the proposed method could be extended to incorporate stitching, or to discuss its potential limitations in scenarios where stitching errors are significant.

**Paper Formatting Concerns:**

The reference format is not consistent.

**Quality:**

3

**Strengths And Weaknesses:**

## Strengths

- The paper presents a well-motivated and clearly explained workflow for global trajectory optimization in EM 3D registration. The use of Neural ODEs and the dual-objective formulation are conceptually sound and well-supported by the mathematical derivation.
- Experimental results across multiple datasets show strong performance compared to existing methods, and the theoretical exposition aids in understanding the model's underlying mechanisms.

## Weaknesses

- Several figures lack sufficient clarity. For example, in Figure 1(b), it is unclear what the “previous method” refers to, what the color-coded outputs represent, and which dataset is used. Figure 3(a) is not referenced or discussed in the main text, leaving the role of grid sampling and bilinear splatting unexplained.
- The ablation study is limited to a single dataset (EFPL), which restricts the generalizability of the analysis.
- Runtime comparisons are missing—an important concern for large-scale EM users.
- The paper lacks discussion of failure cases or scenarios where the method may struggle, such as with highly noisy data. While the framework is said to incorporate biophysical priors, the modeling and validation of these priors remain vague. Recent deep learning-based methods for dense optical flow or registration are not mentioned in the comparisons. Finally, the scalability and memory demands of pixel-wise trajectory tracking across large volumes are not addressed, which raises concerns for TB-scale ssEM datasets.

---

> ### Author Rebuttal · Authors · 2025-07-30
>
> #### **Weaknesses 1**
> Thank you for the question. The “previous works” in Figure 1 refer to pairwise registration methods. The demo in Figure 1 uses the EFPL dataset, and the color-coded represents deformation field. We apologize for the missing reference to Figure 3(a), which is intended to contrast grid interpolation and splatting. We will clarify these figures with more explicit explanations in the revision.
>
> #### **Weaknesses 2**
> Thank you for the suggestion. We have extended our ablation studies to four additional datasets, as presented in Tables 8–11 below. The consistent performance patterns observed across all datasets demonstrate the generalizability and robustness of our method beyond a single test set. In addition, we have included several other ablation experiments in response to suggestions from other reviewers (see our responses to Cons 2 from Reviewer L5DT). We will include all results and discussion in the revised version.
>
> **Table 8**
> ||TS Loss term|DF Loss term|SC Loss term|SSIM|
> |-|-|-|-|-|
> |Mus Heart|✓|✗|✗|0.721|
> ||✓|✓|✗|0.855|
> ||✓|✗|✓|0.793|
> ||✓|✓|✓|0.872|
> |Mus Kidney|✓|✗|✗|0.698|
> ||✓|✓|✗|0.822|
> ||✓|✗|✓|0.774|
> ||✓|✓|✓|0.851|
>
> **Table9**
> ||SSIM|0.0001|0.001|0.01|0.1|
> |-|-|-|-|-|-|
> |MusHeart|0.001|0.775|0.782|0.771|0.762|
> ||0.01|0.795|0.806|0.783|0.779|
> ||0.1|0.852|0.872|0.841|0.822|
> ||1|0.811|0.819|0.805|0.787|
> |MusKidney|0.001|0.754|0.762|0.752|0.741|
> ||0.01|0.781|0.794|0.772|0.766|
> ||0.1|0.833|0.851|0.821|0.815|
> ||1|0.768|0.773|0.758|0.743|
>
> **Table10**
> |SSIM\Iterations|50|100|150|200|250|300|
> |-|-|-|-|-|-|-|
> |MusHeart|0.719|0.802|0.855|0.872|0.874|0.874|
> |MusKidney|0.713|0.796|0.828|0.851|0.851|0.851|
> |MusLiver|0.695|0.774|0.811|0.842|0.844|0.844|
> |MusSkin|0.682|0.788|0.842|0.864|0.868|0.866|
>
> **Table 11**
> ||Gauss-Seidel|NeuralODEs|SSIM|
> |-|-|-|-|
> |Mus Heart|✔|✘|0.847|
> ||✘|✔|0.835|
> ||✔|✔|0.872|
> |Mus Kidney|✔|✘|0.833|
> ||✘|✔|0.827|
> ||✔|✔|0.851|
> |Mus Liver|✔|✘|0.816|
> ||✘|✔|0.803|
> ||✔|✔|0.842|
> |Mus Skin|✔|✘|0.843|
> ||✘|✔|0.825|
> ||✔|✔|0.864|
>
> #### **Weaknesses 3**
> Thank you for your question. As reported in Table 2 of the Supplementary Materials, our method achieves the best accuracy with an SSIM of 0.87, though the runtime is 2.46s. To further improve efficiency, we have now added new experiments focused on optimization strategies. We significantly reduce the runtime (from 2.46s to 0.95s) while maintaining SOTA-level accuracy. This makes our approach more applicable for large-scale EM use cases. Please refer to our response to Cons3 from Reviewer L5DT for detailed experimental results and analysis.
>
> #### **Weaknesses 4**
> Thank you for your comments. We addressed challenging scenarios like high noise and nonlinear deformations by: 1) data augmentation by adding noise during training and 2) upgrading the baseline tracker with advanced optical flow networks (see our response to Q.2 from Reviewer Luv5). The results, e.g., on the mouse heart dataset with 10% noise, demonstrate that both strategies significantly improve performance: data augmentation improves SSIM from 0.832 to 0.852, and upgrading the tracking network further boosts it to 0.883.
>
> For varying nonlinear deformations, Table 12 shows our method consistently outperforms GaussReg (e.g., SSIM on mouse heart drops from 0.832 to 0.742 for GaussReg, but only from 0.872 to 0.803 for ours).
>
> Our biological prior—based on the smooth manifold nature of structures—introduces temporal smoothness and spatial consistency losses (see Line 140 of the main text) to ensure plausible deformations. Ablation studies in Table 2  of the main text and additional datasets beyond EFPL (see our above response to Weaknesses 2) confirm the effectiveness and robustness of this design.
>
> We agree that using a stronger tracking network improves results, with advanced optical flow models boosting SSIM by 0.3–0.5 (see our response to Cons 2 from  Reviewer L5DT). However, our work presents a novel paradigm, not just an incremental upgrade (as noted by Reviewer TWfx). A simple network already demonstrates great potential of our framework, though it leaves room for further improvements with better tracking components.
>
> We optimized the speed–accuracy trade-off for large-scale EM data, reducing runtime from 2.46s to 0.95s while maintaining SOTA SSIM accuracy (see our response to Cons 3 from Reviewer L5DT). Scalability and memory for TB-scale datasets are also addressed (see our below response to Q.4). All results will be included in the revised version.
>
> **Table 12**
> || GaussReg (α=1.0) | Ours (α=1.0) | GaussReg (α=1.5) | Ours (α=1.5) | GaussReg (α=2.0) | Ours (α=2.0) | GaussReg (α=2.5) | Ours (α=2.5) | GaussReg (α=3.0) | Ours (α=3.0) |
> |-|-|-|-|-|-|-|-|-|-|-|
> |Mus Heart|0.832|0.872|0.811|0.867|0.772|0.856|0.753|0.834|0.742|0.803|
> |Mus Kidney|0.781|0.851|0.763|0.843|0.739|0.823|0.725|0.809|0.702|0.776|
> |Mus Liver|0.805|0.842|0.782|0.836|0.758|0.817|0.733|0.782|0.721|0.762|
> |Mus Skin|0.827|0.864|0.797|0.851|0.763|0.838|0.746|0.821|0.737|0.79|
>
> #### **Q.1**
> Thank you for raising this important question. For the synthetic experiments, we adopt the high-quality volumes from the OpenOrganelle dataset as the ground-truth references. These volumes are acquired with isotropic resolution and minimal distortion, making them suitable for constructing controlled benchmarks.
>
> To simulate realistic z-axis deformations, we synthetically warp the volume along the axial direction using smooth deformation fields generated by randomly sampling Gaussian noise fields (see Eq. (1) in the Supplementary Materials). Independent perturbations are applied to each slice (except for the first) to introduce realistic inter-slice misalignments (see Section A2.1 for details in the Supplementary Materials).
>
> To ensure the realism of the synthetic distortions, we strictly follow the deformation protocols established in widely used baselines such as GaussReg, EFSR, and EMReg, which introduce slice-wise perturbations that closely mimic real-world misalignments observed in ssEM data. This benchmark design enables controlled evaluation while preserving biological plausibility.
>
> #### **Q.2**
> Thank you for the question. To the best of our knowledge, EMReg, as presented in the original paper, focuses solely on pairwise registration, and neither the method nor the reported results extend to full-stack or global 3D registration. Therefore, EMReg has not been adapted to optimize over larger blocks with global regularization. If the reviewer is aware of subsequent work where EMReg has been applied for global regularization across larger volumes, we would be happy to analyze and compare accordingly.
>
> Regarding baseline implementations, we used the official EMReg implementation and extended it to 3D registration via sequential pairwise registration. For TrakEM2, we employed the FIJI “Elastically Align Stack” plugin. For SEAMLeSS, we used the official Corgie package for 3D registration. For EFSR and GaussReg, we followed the settings of the original papers. All baselines were trained on the same data as our method, with hyperparameters set according to their respective papers. We will clarify all baseline configurations and resolve citation inconsistencies in the revised version.
>
> #### **Q.3**
> We appreciate your mention of SOFIMA, whose distributed block processing strategy has been inspiring to us. After carefully reviewing its source code, we found that it employs an "elastic mesh optimizer" for registration but acknowledges that the optimization proceeds sequentially, section by section. This approach differs fundamentally from our design of global trajectory optimization.As Reviewer TWfx noted, we conceptually redefine registration as a global trajectory optimization problem—an elegant and significant contribution that directly addresses the fundamental issue of error accumulation inherent in pairwise methods. Due to the time constraint for rebuttal, we will discuss this in the revised version.
>
> #### **Q.4**
> Thank you for the suggestion. We optimized the speed–accuracy trade-off, reducing runtime from 2.46s to 0.95s while maintaining SOTA SSIM accuracy (see our response to Cons 3 from Reviewer L5DT). Our framework can scale to TB-scale ssEM datasets using a parallel, block-wise strategy like SOFIMA: 1) split volumes into chunks; 2) run parallel trajectory tracking and inversion on multi-GPUs; 3) stitch local deformation fields; 4) apply the global deformation.
>
> As noted in Section A2.3 of Supplementary Materials, for datasets containing thousands of slices, we split them into blocks of 100 slices, which can be processed comfortably on a 40GB GPU. We will detail this scalable design in the revision.
>
> #### **Limitation 1**
> Thank you for your suggestion. We have explored the speed–accuracy trade-off for large-scale EM data. Our optimization significantly reduces runtime (from 2.46s to 0.95s), while preserving SOTA-level SSIM accuracy (see our responses to Cons 3 from Reviewer L5DT).
>
> #### **Limitation 2**
> Thank you for your suggestion. TrakEM2 is a widely used EM image processing toolkit that supports both stitching and registration. However, existing frameworks—including TrakEM2—typically adopt a sequential pipeline that performs stitching first, followed by registration. Our method is also compatible with such pipelines.
>
> But we agree that integrating stitching and registration into a unified framework is a promising direction. A potential solution is to leverage shared encoder features between the stitching and trajectory tracking modules, enabling joint optimization.
>
> Moreover, we agree that stitching errors can affect the trajectory tracking. To assess the impact, we added evaluations under more challenging conditions (see our responses to Q.2 from Reviewer Luv5), and further replaced the tracking module with more advanced networks to improve robustness against artifacts (see our responses to Cons 2 from Reviewer L5DT). We will incorporate all these clarifications in the revised version.

---

> ### Author Response · Authors · 2025-08-05
>
> Dear Reviewer zfxm,
>
> Thank you very much for your time and valuable comments on our work.
>
> As the author-reviewer discussion phase will draw to its end soon, we would greatly appreciate it if you could let us know whether our rebuttal has addressed your concerns, or if any part still requires further clarification.
>
> Best regards,
>
> Authors

---

> > ### Comment · Reviewer_zfxm · 2025-08-06
> >
> > Thanks for the detailed response. My questions have been well addresseed.

---

> > > ### Author Response · Authors · 2025-08-06
> > >
> > > Dear Reviewer zfxm,
> > >
> > > Thank you very much for your kind reply. We're  very glad to hear that our rebuttal has well addressed all of your concerns.
> > >
> > > We will be more than happy to answer your futher questions if any. If you are satisfied with our rebuttal, we would be truly grateful if you could increase your score.
> > >
> > > Best regards,
> > >
> > > Authors

---

### Official Review · Reviewer_Luv5 · 2025-07-02

**Clarity:** 3
**Significance:** 3
**Originality:** 3
**Rating:** 5
**Confidence:** 3

**Summary:**

The authors provide motivation from their application (3D registration of series section electron microscopy) for a 3D registration method that can handle non-linearities. Neural ODEs are a reasonable approach for this. They compare their method to existing methods in the field and demonstrate superior performance under certain metrics.

**Questions:**

* You refer to "biological priors" throughout the paper (including the abstract) but never define exactly what you mean by it or how they are integrated into the model

* How does your method handle noise and imaging artefacts? Pixel tracking methods are sensitive in particular to the former

* Suggested ablation study - what happens if you remove the ODESolver / neural compensator component in Equation 10? Capturing non-linearity is your central argument, it would be instructive to test how important this is to your methodology

L249 - any deeper insight into why your method outperforms GaussReg?

* The results are a bit underwhelming - only in synthetic data, and no comment on the key requirement of handling non-linearities. How does it perform with real data, with noise and non-linearities?

* Gauss-Seidel operator - this is introduced briefly but needs defining better

* Trajectory inversion - why was Gaussian splatting chosen over other interpolation methods?

**Ethical Concerns:**

["NO or VERY MINOR ethics concerns only"]

**Final Justification:**

The authors satisfactorily answered my questions, particularly regarding the novelty of the work over established registration methods.

**Limitations:**

Please see my above comments regarding limitations.

**Paper Formatting Concerns:**

None.

**Quality:**

3

**Strengths And Weaknesses:**

Strengths
- Neural ODEs are a reasonable approach for the task
- There are some interesting ideas in the paper

Weaknesses
- The use of ODE based methods for this task is not novel in itself
- For an application-driven paper, the Results section is a bit underwhelming - only shows synthetic data, and no comment on the key requirement of handling non-linearities. How does it perform with real data, with noise and non-linearities?

---

> ### Author Rebuttal · Authors · 2025-07-30
>
> #### **Weaknesses 1**
> Thank you for raising concerns on novelty. While Neural ODEs have been applied in many other domains, to the best of our knowledge, we are the first to introduce Neural ODEs for 3D registration in ssEM data.
>
> More importantly, our key contribution lies not merely in the use of Neural ODEs, but in the conceptual reformulation of registration as an unsupervised global trajectory optimization problem—an innovation that directly addresses the long-standing issue of error accumulation in traditional pairwise methods (as succinctly summarized by Reviewer TWfx). Neural ODEs are simply a natural and effective mechanism for realizing this paradigm.
>
> #### **Weaknesses 2**
> Thank you for the valuable suggestion. We already showed registration results on four 1024 × 1024 × 1299 subvolumes from different regions of the real FemFlyBrain dataset in Figures 12–13 of our Supplementary Materials. As you can see, our method achieves improved structural continuity and better restoration of correct axial structures compared to baselines. Following your suggestion, we have now additionally evaluated our method on two blocks from the FAFB dataset and four blocks from the mouse cortical dataset (see our responses to Cons 1 from Reviewer L5DT). We will include the corresponding results in the revision.
>
> #### **Q.1**
> Thank you for the valuable question. The biological prior refers to the smooth manifold nature of biological structures, where nonlinear deformation primarily arises from gradual local curvature changes (as mentioned at the end of the third paragraph in Section 1 of the main text).
>
> Based on this prior, we introduce two regularization terms in our trajectory optimization module: a temporal smoothness loss and a spatial consistency loss (see Line 140 in the main text). These are designed to promote biologically plausible deformations.
>
> To validate the correctness of this biological prior modeling, we conduct ablation studies in Table 2 of the main text. In addition, we now include the same ablation experiments on two more datasets beyond EFPL (see our response to Weaknesses 2 from Reviewer zfxm). The results show consistent patterns in peak performance improvements, further supporting the robustness of our prior-based design. We will clarify this point more explicitly in the revised version.
>
> #### **Q.2**
> Thank you for the suggestion. We agree with you that noise and imaging artifacts would affect the trajectory tracking stage. To address this issue, we adopt two strategies: 1) introducing data augmentation during training and 2) replacing the baseline tracker with more advanced optical flow networks. To validate these approaches, we conducted experiments on four datasets with varying noise levels, comparing our method with GaussReg (Table 6).
>
> The results (e.g., on the mouse heart dataset with 10% noise) demonstrate that both strategies significantly improve performance: data augmentation improves SSIM from 0.832 to 0.852, and upgrading the tracking network further boosts it to 0.883. In contrast, GaussReg achieves only 0.788. These results highlight that our method remains robust under noisy conditions. Importantly, this indicates that our work represents a novel paradigm in ssEM registration rather than a mere incremental improvement, with significant room for further gains even using more powerful networks and training tricks.
>
> **Table6**
> ||5%GaussReg|5%Orgin|5%Orgin+DataAug|5%SEARAFT+DataAug|10%GaussReg|10%Orgin|10%Orgin+DataAug|10%SEARAFT+DataAug|15%GaussReg|15%Orgin|15%Orgin+DataAug|15%SEARAFT+DataAug|
> |-|-|-|-|-|-|-|-|-|-|-|-|-|
> |MusHeart|0.815|0.851|0.863|0.896|0.788|0.832|0.852|0.883|0.748|0.771|0.853|0.87|
> |MusKidney|0.762|0.833|0.854|0.915|0.734|0.822|0.847|0.893|0.715|0.759|0.834|0.882|
> |MusLiver|0.779|0.814|0.842|0.895|0.74|0.784|0.845|0.877|0.72|0.714|0.83|0.863|
> |MusSkin|0.804|0.837|0.857|0.922|0.751|0.803|0.824|0.905|0.728|0.742|0.827|0.88|
>
> SEA-RAFT: SEA-RAFT: Simple, Efficient, Accurate RAFT for Optical Flow, ECCV 2024
>
> #### **Q.3**
> Thank you for your suggestion. We provided ablation studies in Table 5 of our Supplementary Materials by removing Neural ODEs. The results show that removing the Neural ODE component leads to performance drop, with SSIM decreasing from 0.746 (full model) to 0.721.
>
> To further validate the importance of this module, we additionally performed the same ablation experiments on multiple datasets beyond EFPL (see our response to Weaknesses 2 from Reviewer zfxm). Consistent patterns were observed across these datasets, confirming that the Neural ODEs are critical to the effectiveness of our method.
>
> #### **Q.4**
> Thank you for the valuable question. GaussReg performs optimization within local slice windows, which inherently limits its ability to maintain global consistency across long-range sections. In contrast, our method adopts a global trajectory-based formulation that enables coherent alignment over large spatial extents. This key advantage is quantitatively supported by the error accumulation scatter plots shown in Figure 4 of the main text and Figures 6–11 of the Supplementary materials, where our method demonstrates reduced error accumulation compared to GaussReg. We will further elaborate on this important distinction in the revision.
>
> #### **Q.5**
> Thank you for your suggestion. Regarding the effectiveness of our method on more real-world data, we already presented registration results on four 1024 × 1024 × 1299 subvolumes from different regions of the real FemFlyBrain dataset in Figures 12–13 of the Supplementary Materials. As you can see, our method demonstrates improved structural continuity and more accurate restoration of axial anatomical structures compared to the baselines. Following your suggestion, in addition to the four real FemFlyBrain blocks included in the supplementary material, we further conduct experiments on six blocks from two widely used real datasets (see our responses to Cons 1 from Reviewer L5DT), we will report the corresponding results in the revised version.
>
> #### **Q.6**
> Thank you for the valuable suggestion. We employ the Gauss-Seidel method, a classical iterative solver for linear systems $A\mathbf{x} = \mathbf{b}$ that leverages the most recent updates to accelerate convergence, particularly in sparse, tridiagonal systems.
>
> Our smoothing objective (Eq. (2) in Section A.3 of Supplementary Materials) is:
>
> $$
> E(\mathcal{P}) = \sum_{z=1}^{N-1} \left\| \mathcal{P}(z{-}1) - 2\mathcal{P}(z) + \mathcal{P}(z{+}1) \right\|^2 + \lambda \sum_{z=1}^{N-1} \left\| \mathcal{P}(z) - \mathcal{P}^{(0)}(z) \right\|^2.
> $$
>
> As derived in Section A.3 of Supplementary Materials, minimizing this energy leads to the following linear system:
>
> $$
> (4 + \lambda)\mathcal{P}(z) - 2\mathcal{P}(z{-}1) - 2\mathcal{P}(z{+}1) = \lambda \mathcal{P}^{(0)}(z), \quad z = 1, \ldots, N{-}1.
> $$
>
> This system can be equivalently written as a symmetric tridiagonal linear system:
>
> $$
> A \mathbf{p} = \mathbf{b},
> $$
>
> where:
>
> * $\mathbf{p} = [\mathcal{P}(1), \ldots, \mathcal{P}(N{-}1)]^\top$ is the unknown trajectory,
> * $\mathbf{b} = [\lambda \mathcal{P}^{(0)}(1), \ldots, \lambda \mathcal{P}^{(0)}(N{-}1)]^\top$,
> * and $A \in \mathbb{R}^{(N{-}1) \times (N{-}1)}$ is a tridiagonal matrix with structure:
>
> $$
> A =
> \begin{bmatrix}
> 4+\lambda & -2         &           &        &  \\
> -2        & 4+\lambda & -2         &        &  \\
>           & \ddots    & \ddots     & \ddots &  \\
>           &           & -2         & 4+\lambda & -2 \\
>           &           &            & -2     & 4+\lambda
> \end{bmatrix}.
> $$
>
> Applying the Gauss-Seidel method to this system, the general update rule for the $i$-th variable is:
>
> $$
> x_i^{(k+1)} = \frac{1}{A_{ii}} \left( b_i - \sum_{j<i} A_{ij} x_j^{(k+1)} - \sum_{j>i} A_{ij} x_j^{(k)} \right).
> $$
>
> For our tridiagonal matrix, each row $z$ only has nonzero entries at $z-1$, $z$, and $z+1$. Substituting these into the general Gauss-Seidel formula, the update rule for $\mathcal{P}(z)$ becomes:
>
> $$
> \mathcal{P}^{(k+1)}(z) = \frac{1}{A_{zz}} \left( \lambda \mathcal{P}^{(0)}(z) - A_{z,z-1} \mathcal{P}^{(k+1)}(z{-}1) - A_{z,z+1} \mathcal{P}^{(k)}(z{+}1) \right),
> $$
>
> where:
>
> * $A_{zz} = 4 + \lambda$,
> * $A_{z,z-1} = A_{z,z+1} = -2$,
> * and we use the latest available updates for $\mathcal{P}^{(k+1)}(z{-}1)$, along with the previous values for $\mathcal{P}^{(k)}(z{+}1)$.
>
> Substituting these in yields:
>
> $$
> \mathcal{P}^{(k+1)}(z) = \frac{1}{4 + \lambda} \left( 2 \mathcal{P}^{(k+1)}(z{-}1) + 2 \mathcal{P}^{(k)}(z{+}1) + \lambda \mathcal{P}^{(0)}(z) \right).
> $$
>
> For simplicity, we divide both numerator and denominator by 2, resulting in the final explicit Gauss-Seidel update:
>
> $$
> \mathcal{P}^{(k+1)}(z) = \frac{\mathcal{P}^{(k+1)}(z{-}1) + \mathcal{P}^{(k)}(z{+}1) + \lambda \mathcal{P}^{(0)}(z)}{2 + \lambda}, \quad z = 1, \ldots, N{-}1.
> $$
>
> This completes the derivation of the iterative update rule used for trajectory smoothing, as presented in Eq.8 of the main text.
>
> #### **Q.7**
> Thank you for the valuable question. We adopt bilinear splatting (not Gaussian splatting) in the inversion step due to its favorable speed–accuracy trade-off. To validate this choice, we now provide a comparison among bilinear, trilinear, and classical radial basis function (RBF) interpolation in Table 7.
>
> The results show that bilinear interpolation achieves an SSIM of 0.87 within only 0.008 seconds. In contrast, classical RBF interpolation, even when accelerated by GPU, takes 1.2 seconds but yields only marginal improvement (SSIM = 0.89). While learning-based interpolation methods may potentially yield further improvements, we leave this for future work due to time constraints for rebuttal.
>
> **Table7**
> |Method|Time(s)|SSIM|
> |-|-|-|
> |Bilinear splatting|0.008|0.87|
> |Trilinear splatting|0.0056|0.87|
> |RBF(CPU)|21.4|0.89|
> |RBF(GPU)|1.2|0.89|

---

> ### Author Response · Authors · 2025-08-05
>
> Dear Reviewer Luv5,
>
> Thank you very much for your time and valuable comments on our work.
>
> As the author-reviewer discussion phase will draw to its end soon, we would greatly appreciate it if you could let us know whether our rebuttal has addressed your concerns, or if any part still requires further clarification.
>
> Best regards,
>
> Authors

---

> > ### Comment · Reviewer_Luv5 · 2025-08-05
> >
> > Thank you for your detailed responses. My questions are answered and I will increase my score accordingly.

---

> > > ### Author Response · Authors · 2025-08-05
> > >
> > > Dear Reviewer Luv5
> > >
> > > Thank you sincerely for your valuable time, your positive assessment of our work, and for increasing your score.
> > >
> > > We promise to fully address all of your concerns in the revised version.
> > >
> > > Could you please let us know if there are any other concerns that still remain? We would be more than happy to address them and revise further to improve the quality of this work. Once again, we truly appreciate your thoughtful and constructive feedback.
> > >
> > > Best regards,
> > >
> > > Authors

---

### Official Review · Reviewer_L5DT · 2025-07-03

**Clarity:** 3
**Significance:** 3
**Originality:** 3
**Rating:** 4
**Confidence:** 5

**Summary:**

1. This paper first analyses the intrinsic features of ssEM registration problem, and defines 3D registration as a continuous manifold trajectory optimization problem.
2. This paper proposes a new unsupervised learning method by integrating biophysical priors with data-driven deformation compensation.
3. This paper conducts the synthetic and realistic experiments to verify the effectiveness of the proposed method.

**Questions:**

Please refer to "cons" in Strengths And Weaknesses

1. Does the proposed method perform better than other methods, especially for the traditional methods, on the unseen ssEM datasets? In the experiment, the main results focus on the synthetic datasets, and EFPL dataset is used for the ablation studies. And the real-world experiments in the supplementary materials also show the training and testing results? I wonder about the generalizable performance of the proposed method with unsupervised learning schemes.
2. The ablation experiments on the trajectory tracking module are required. Do the hyperparameters in Eq.15 influence the final performance obviously? And the powerful tracking network, such as more parameters or advanced networks, can give better performance?
3. The efficiency problem is important. And how do you use the proposed method for the large-scale long-slices ssEM dataset? Can the method save the inference burden by selecting several trajectories rather than considering all pixels?
4, Dice score and Folds are demanded for the performance comparison. Can the author evaluate the neuron segmentation smoothness and deformation smoothness?

**Ethical Concerns:**

["NO or VERY MINOR ethics concerns only"]

**Final Justification:**

In the rebuttal period, authors give detailed explanations and results to address my concerns.

For the large-scale EM registration results, the authors emphasise the results in the appendix and promise to supplement more results in the final version.

Meanwhile, other concerns, such as more metrics (Dice and Folds) and better tracking backbones, are discussed and solved to some extent.

As a result, I will change my score from "Borderline reject" to "Borderline accept".

**Limitations:**

Please refer to "Questions" and "Weaknesses".

**Paper Formatting Concerns:**

I think the paper is easy to follow, and no obvious formatting concerns are noticed.

**Quality:**

2

**Strengths And Weaknesses:**

Pros:
1. The motivation is straightforward and the idea is reasonable. The global optimization along the slice dimension is a solution to the ssEM registration problem, which shares a similar core idea with some traditional methods
2. The detailed operations in the method make sense and are easy to follow.
3. The proposed method shows superior performance compared with other comparison methods.


Cons:
1. The main experiments focus on the synthetic dataset and lack comprehensive experiments on the real-world/large-scale ssEM datasets.
2. The effect of the trajectory tracking module is clear. Accurate tracking is an important starting point for the following optimization, but the current discussion is not enough.
3. The efficiency problem should be considered. Compared with GaussReg, the SSIM improves from 0.82 to 0.87, but the utilized time also improves from 0.92 to 2.46. Though authors claim the better performance compared with GaussReg, more situations without labels should be conducted and analyzed.
4. More metrics should be utilized, such as Dice score in some labelled ssEM dataset, which can better reflect the instance coherence.

---

> ### Author Rebuttal · Authors · 2025-07-30
>
> #### **Cons 1**
> Thank you for the valuable suggestion. We already showed registration results on four $1024 \times 1024 \times 1299$ subvolumes from different regions of the real FemFlyBrain dataset in Figures 12–13 of our Supplementary Materials. As you can see, our method achieves improved structural continuity and better restoration of correct axial structures compared to baselines.
>
> Following your suggestion, we have now additionally evaluated our method on two $1024 \times 1024 \times 700$ blocks from the FAFB dataset [1] and four $1024 \times 1024 \times 500$ blocks from the mouse cortical dataset [2]. Due to the lack of established quantitative metrics for this task on real data and the restriction of not being able to include figures in rebuttal, we will include corresponding results and visual comparisons in the revised version.
>
> [1] A complete electron microscopy volume of the brain of adult *Drosophila melanogaster*, *Cell*, 2018.
> [2] Saturated reconstruction of a volume of neocortex, *Cell*, 2015.
>
> #### **Cons 2**
>
> We agree with the reviewer that accurate tracking is a critical foundation for the subsequent optimization stages. In response, we have now added ablation studies on the trajectory module, including:
> (i) replacing it with a stronger dense tracking network (Table 1), and
> (ii) a sensitivity analysis of the key hyperparameter $\lambda$ (Table 2).
>
> To ensure generality, these experiments are also conducted on datasets beyond EFPL. Table 1 shows that incorporating more advanced optical flow models can further improve the performance, yielding 0.3--0.5 higher SSIM scores. Table 2 demonstrates that our method remains robust and maintains SOTA-level accuracy across a wide range of $\lambda$ values (between 1.5 and 7.5).
>
> We would like to emphasize that our work introduces a novel paradigm rather than merely an incremental improvement (as aptly noted by Reviewer TWfx). To highlight this, we intentionally adopted a simple baseline network architecture in the original manuscript, which leaves significant room for further performance gains by integrating more sophisticated tracking components.  All the additional results will be included in the revised version.
>
> **Table 1.**
> | SSIM| RAFT (ECCV 2020) | GAFlow (ICCV 2023) | SEA-RAFT (ECCV 2024) |
> |-|-|-|-|
> | Mus Heart|0.892| 0.917| 0.909|
> | Mus Kidney|0.894| 0.933| 0.925|
> | Mus Liver| 0.871| 0.928|0.918 |
> | Mus Skin| 0.920| 0.943|0.932|
>
> RAFT: Raft: Recurrent all-pairs field transforms for optical flow. ECCV, 2020
>
> GAFlow: GAFlow: Incorporating Gaussian Attention into Optical Flow, ICCV, 2023
>
> SEA-RAFT: SEA-RAFT: Simple, Efficient, Accurate RAFT for Optical Flow, ECCV 2024
>
> **Table 2.**
> | SSIM \| $\lambda$ |0.15|1.5|4.5|7.5|10|
> |-|-|-|-|-|-|
> |Mus Heart|0.855|0.872|0.870|0.864| 0.830|
> |Mus Kidney|0.827|0.851|0.855|0.843| 0.809|
>
> #### **Cons 3**
>
> Thank you for the insightful suggestion regarding runtime efficiency. In response, we have added new experiments focused on efficiency optimization through:
> (i) sparse trajectory sampling (every 2/3/4 pixels) (**Table 3**), and
> (ii) resolution reduction via down-sampling followed by up-sampling (**Table 4**).
>
> **Table 3** shows that sparse sampling significantly reduces runtime (from 2.46s to 0.95s), while preserving SOTA-level accuracy, with SSIM only decreasing slightly from 0.874 to 0.831.
> **Table 4** demonstrates that reducing resolution also improves speed (from 2.46s to 0.91s), with a drop in accuracy (SSIM from 0.874 to 0.827), which still outperforms the GaussReg baseline. The observed performance drop may be attributed to the use of a simple bilinear up-sampling strategy. We believe that combining sparse trajectory sampling with resolution reduction via down-sampling followed by up-sampling can achieve a better trade-off between runtime and accuracy.We will include the above results and further discussion in the revised version.
>
> For more situations without labels, we also include more experiments on real datasets and compare with GaussReg (see our response to Cons 1 above).  We will include the complete results and extended discussion in the revised version.
>
> **Table 3.**
> |               | All pixels | Every 2nd pixel | Every 3rd pixel | Every 4th pixel |
> |---------------|------------|-----------------|-----------------|-----------------|
> | **SSIM**      | 0.874      | 0.866           | 0.842           | 0.831           |
> | **Time (s)**  | 2.46       | 1.35            | 1.12            | 0.95            |
>
> **Table 4.**
> |               | Full resolution | 1/2 resolution | 1/4 resolution |
> |---------------|-----------------|----------------|----------------|
> | **SSIM**      | 0.874           | 0.845          | 0.827          |
> | **Time (s)**  | 2.46            | 1.12           | 0.91           |
>
>
> #### **Cons 4**
> We appreciate your valuable suggestion. As reported in Table 3 of our Supplementary Materials, we have already provided Dice scores for quantitative evaluation. In accordance with your request, we additionally report the Folds metric in Table 5 to assess the diffeomorphic property of the deformation field. The results show that our method achieves the lowest Folds values, consistently around 0.01, whereas most baseline methods report values above 0.1. This indicates significantly fewer grid foldings and better preservation of topological consistency in the deformation fields produced by our approach.
>
> **Table 5.**
> | Folds       | TrakEM2 | EFSR  | EMReg | SEMLeSS | GaussReg | Ours   |
> |-------------|---------|-------|-------|---------|----------|--------|
> | Mus Heart   | /       | 0.188 | 0.243 | 0.135   | 0.099    | 0.0171 |
> | Mus Kidney  | /       | 0.206 | 0.379 | 0.24    | 0.174    | 0.0098 |
> | Mus Liver   | /       | 0.193 | 0.31  | 0.155   | 0.106    | 0.0124 |
> | Mus Skin    | /       | 0.171 | 0.252 | 0.102   | 0.091    | 0.0095 |
>
> #### **Q.1**
>
> Thank you for your valuable suggestion. As stated in the main text (Table 1), all six test datasets are unseen ssEM datasets. We train a single model on training data and evaluate it across these six unseen datasets to demonstrate the generalization ability of our method. Compared to traditional methods TrakEM2, our approach achieves significantly improved performance, for instance, on the mouse heart dataset, the SSIM improves from 0.57 to 0.87.
>
> For real-world datasets, we also adopt a similar protocol: training on a small region and testing on separate, unseen regions (as described in Line 110, Section A2.3 of our Supplementary Materials). The visualizations in Figures 12–13 of Supplementary Materials demonstrate that our method better restores the correct axial structure and generalizes well to new regions that were not observed during training.
>
> Following your suggestion, we additionally include experiments on 6 blocks from two more real-world ssEM datasets (see our response to Cons 1 above). We also provide ablation studies on datasets beyond EFPL (see our response to Weaknesses 2 from Reviewer zfxm), further validating the robustness and generalizability of our framework.
>
> #### **Q.2**
>
> Thank you for your helpful suggestion. We now provide a sensitivity analysis of the key hyperparameter λ in Table 2. The results show that our method remains robust and maintains SOTA-level accuracy across a wide range of λ values (from 1.5 to 7.5).
>
> In addition, regarding the tracking module, we replace the original 2D Unet network with more advanced optical flow models: RAFT (5.3M parameters), GAFlow (10.1M), and SEA-RAFT (4.9M). Table 1 shows that incorporating these more powerful tracking networks can further boost performance, achieving 0.3–0.5 improvements in SSIM scores. Please also refer to our response to Cons 2 above for additional detailed results and discussions.
>
> #### **Q.3**
>
> Thank you for your question. For large-scale long-slice ssEM datasets, we adopt a divide-and-conquer strategy by splitting large volumes into manageable blocks. Our method estimates deformation fields in parallel for each block and then fuses them to reconstruct the full field. Please refer to our response to Q.4 from Reviewer zfxm for more details.
>
> For handling long slices, we process them in blocks (e.g., 100 slices per batch), as described in Line 113, Section A2.3 of our Supplementary Materials.
>
> To address the inference burden, we introduce two efficient strategies as discussed in our response to Cons 3 above: 1) sparse trajectory sampling, and 2) downsample-then-upsample resolution. Experiments show that these strategies significantly reduce inference time from 2.46s to 0.95s, while preserving SOTA accuracy.
>
> #### **Q.4**
>
> We appreciate your valuable suggestion. In Table 3 of our Supplementary Materials, we have already provided Dice scores for quantitative evaluation.
>
> In accordance with your request, we additionally report the Folds metric in Table 5 above to assess the diffeomorphic property of the deformation field. The results show that our method achieves the lowest Folds values, consistently around 0.01, whereas most baseline methods report values above 0.1. This indicates significantly fewer grid foldings and better preservation of topological consistency in the deformation fields produced by our approach.

---

> > ### Comment · Reviewer_L5DT · 2025-08-04
> >
> > Thanks for your detailed rebuttals, and most of my concerns are solved.
> >
> > I hope the real large-scale EM data registration results and related codes will be provided in the final version.
> >
> > I will change my score from "Borderline reject" to "Borderline accept".

---

> > > ### Author Response · Authors · 2025-08-04
> > >
> > > Dear Reviewer L5DT,
> > >
> > > Thank you sincerely for your valuable time, your positive assessment of our work, and for increasing your score.
> > >
> > > We promise that real large-scale EM data registration results will be included in the revised version and the corresponding code will be released. We will also address all of your concerns in full in the revision.
> > >
> > > Could you please let us know if there are any other concerns that still remain? We would be more than happy to address them and revise further to improve the quality of this work. Once again, we truly appreciate your thoughtful and constructive feedback.
> > >
> > > Best regards,
> > >
> > > Authors

---

> > > > ### Comment · Reviewer_L5DT · 2025-08-09
> > > >
> > > > Thanks for your responses, and I do not have other concerns.
> > > >
> > > > One more suggestion, I think the related work about medical image registration, such as 2023TMI-SDHNet, 2024CVPR-CorrMLP, 2022JBHI-RDN, 2024CVPR-HViT, and more works, should be supplemented in the final version.
> > > >
> > > > Best regards,
> > > >
> > > > Reviewer.

---

> > > > > ### Author Response · Authors · 2025-08-09
> > > > > **Official Comment by Authors**
> > > > >
> > > > > Dear Reviewer L5DT,
> > > > >
> > > > > Thank you very much for your valuable time devoted to our discussion and constructive suggestions that help improve our work. We will follow your suggestion to add discussions on these related work to the supplementary materials in the final version.
> > > > >
> > > > > Best regards,
> > > > >
> > > > > Authors

---

### Note · Authors · 2025-08-12

We sincerely thank the AC and all reviewers for their time, effort, and constructive feedback. To save the AC’s time and reinforce the positive consensus, we briefly summarize the key points from the rebuttal and subsequent responses.

All four reviewers—zfxm, TWfx, L5DT and Luv5—found our rebuttal detailed and agreed that it addressed all of their concerns. Reviewers zfxm, L5DT and Luv5 expressed such satisfaction that they had no further questions. Reviewer TWfx “appreciates the authors’ detailed rebuttal” and noted that “the authors have addressed key weaknesses,” while encouraging deeper discussion on deployment practicality, scalability, and clarity of writing.

We fully agree that these are important and valuable directions, and we have addressed them thoroughly in our responses. We believe such explorations are most meaningful after establishing and validating a new paradigm for ssEM 3D registration—which is precisely the focus and core contribution of our work. Our paper presents a paradigm-shifting optimization framework, rather than an incremental improvement over prior methods.

We sincerely hope that our novel perspective and framework will be recognized as a meaningful step beyond existing paradigms. We also fully respect the reviewers’ concerns regarding deployment and scalability, and we believe these will guide valuable future work toward practical large-scale applications. We will incorporate the content of the rebuttal into the revised version. Once again, we thank the reviewers for their valuable suggestions, which have significantly improved the quality of our paper.

---

### Decision · Program_Chairs · 2025-09-17

**Decision:**

Accept (poster)

**Comment:**

This is an application-driven paper that addresses registration in 3D ssEM. To mitigate error accumulation, the authors frame registration as a trajectory optimization problem and design an unsupervised approach to solve it. Reviewers initially criticized the lack of realistic large-scale evidence beyond synthetic data; the clarity of definitions (e.g., “biological priors”), figures, and formatting; and the depth of analysis (scalability, hyperparameters, convergence, …). In rebuttal, the authors effectively addressed many of these points, including adding (or pointing to) results on FemFlyBrain, FAFB, and mouse cortex, adding ablations, reporting additional metrics, and analyzing runtime and various trade-offs. My impression is that this is a solid applied contribution to NeurIPS, of interest in particular to the ssEM community, but also more generally to the ML for microscopy community.

For the camera-ready version, the authors promised to consolidate real-data metrics, runtime and memory tables, sensitivity to the various hyperparameters, and improved figures and definitions. I hope the promised experiments and clarifications are indeed included in the final version.